# Nutrition, Physical Activity, and Dietary Supplementation to Prevent Bone Mineral Density Loss: A Food Pyramid

**DOI:** 10.3390/nu14010074

**Published:** 2021-12-24

**Authors:** Mariangela Rondanelli, Milena Anna Faliva, Gaetan Claude Barrile, Alessandro Cavioni, Francesca Mansueto, Giuseppe Mazzola, Letizia Oberto, Zaira Patelli, Martina Pirola, Alice Tartara, Antonella Riva, Giovanna Petrangolini, Gabriella Peroni

**Affiliations:** 1IRCCS Mondino Foundation, 27100 Pavia, Italy; mariangela.rondanelli@unipv.it; 2Department of Public Health, Experimental and Forensic Medicine, University of Pavia, 27100 Pavia, Italy; 3Endocrinology and Nutrition Unit, Azienda di Servizi alla Persona “Istituto Santa Margherita”, University of Pavia, 27100 Pavia, Italy; milena.faliva@gmail.com (M.A.F.); gaetanclaude.barrile01@universitadipavia.it (G.C.B.); alessandro.cavioni01@universitadipavia.it (A.C.); francesca.mansueto01@universitadipavia.it (F.M.); giuseppe.mazzola02@universitadipavia.it (G.M.); letizia.oberto@gmail.com (L.O.); zaira.patelli01@universitadipavia.it (Z.P.); martina.pirola03@universitadipavia.it (M.P.); alice.tartara01@universitadipavia.it (A.T.); 4Research and Development Department, Indena SpA, 20139 Milan, Italy; antonella.riva@indena.com (A.R.); giovanna.petrangolini@indena.com (G.P.)

**Keywords:** bone mineral density, osteopenia, osteoporosis, nutrition, food pyramid

## Abstract

Bone is a nutritionally modulated tissue. Given this background, aim of this review is to evaluate the latest data regarding ideal dietary approach in order to reduce bone mineral density loss and to construct a food pyramid that allows osteopenia/osteoporosis patients to easily figure out what to eat. The pyramid shows that carbohydrates should be consumed every day (3 portions of whole grains), together with fruits and vegetables (5 portions; orange-colored fruits and vegetables and green leafy vegetables are to be preferred), light yogurt (125 mL), skim milk (200 mL,) extra virgin olive oil (almost 20 mg/day), and calcium water (almost 1 l/day); weekly portions should include fish (4 portions), white meat (3 portions), legumes (2 portions), eggs (2 portions), cheeses (2 portions), and red or processed meats (once/week). At the top of the pyramid, there are two pennants: one green means that osteopenia/osteoporosis subjects need some personalized supplementation (if daily requirements cannot be satisfied through diet, calcium, vitamin D, boron, omega 3, and isoflavones supplementation could be an effective strategy with a great benefit/cost ratio), and one red means that there are some foods that are banned (salt, sugar, inorganic phosphate additives). Finally, three to four times per week of 30–40 min of aerobic and resistance exercises must be performed.

## 1. Introduction

Bones are characterized by a dynamic structure [1] continuously resorbed and rebuilt by osteoclasts and osteoblasts, respectively.

The balance between bone resorption and bone formation and its regulation are critical factors for maintaining adequate mineral homeostasis and bone density. Osteoporosis is caused by an impaired balance of these two remodeling processes, resulting in more bone resorption than bone deposition. There are several clinical conditions that can lead to an imbalance in this remodeling process: old age and postmenopausal period are the major causes of osteoporosis, but other risk factors, including medications, endocrine disorders, immobilization, inflammatory arthropathy, hematopoietic disorders, and nutrition disorders, can also be involved.

Osteoporosis has been defined by The World Health Organization (WHO) as: “the osteoporotic state of the bone is defined by the T-score variable, which is the number of standard deviations (SDs) by which a patient’s test differs from the mean of the young adult reference group (positive t-score values are associated with greater bone mass density than the reference group, negative ones with less bone density values)” [2].

Osteopenia is considered as a state of lower-than-average bone density resulting in an intermediate risk similar to other clinical entities, such as prehypertension, impaired fasting glucose, and borderline high cholesterol [3].

The major complication of osteopenia/osteoporosis is an increase in both traumatic and fragility fractures [4,5] leading to morbidity, mortality, and decreased quality of life (depression, physical disability, loss of independence, and premature death) [6].

Lifestyle measures, such as healthy and balanced diet and physical activity, begun in childhood and carried on throughout life, are factors that are well known as associated with better growth and aging of the bone [7,8,9,10].

A balanced nutritional intake of specific nutrients (from a specific diet and/or through a dietary integration) can be considered as the first step for an effective preventive strategy in an osteopenic state [11].

Given this background, the aim of this narrative review is to evaluate the evidence regarding the ideal dietary therapy in order to reduce the risk of loss of bone mineral density (BMD) and to construct a food pyramid for osteopenic/osteoporotic subjects.

## 2. Materials and Methods

This narrative review was performed following these steps [12]: (1) Configuration of a working group: three operators skilled in clinical nutrition (one acting as a methodological operator and two participating as clinical operators); (2) formulation of the revision question on the basis of considerations made in the abstract: “the state of the art on ideal dietary therapy in order to reduce the risk of loss of bone mineral density (BMD);” (3) identification of relevant studies: a research strategy was planned on PubMed (Public MEDLINE run by the National Center of Biotechnology Information (NCBI) of the National Library of Medicine of Bethesda (Bethesda, MD, USA)) as follows: (a) definition of the keywords (bone mineral density, foods, nutrients, dietary supplement, osteopenia, osteoporosis), allowing the definition of the interest field of the documents to be searched, grouped in inverted commas (“. . .”) and used separately or in combination; (b) use of the Boolean (a data type with only two possible values: true or false) AND operator, which allows the establishments of logical relations among concepts; (c) research modalities: advanced search; (d) limits: time limits: papers published in the last 30 years; humans; languages: English; (e) manual search performed by senior researchers experienced in clinical nutrition through the revision of reviews and individual articles on ideal dietary therapy in order to reduce the risk of loss of BMD published in journals qualified in the Index Medicus; (4) analysis and presentation of the outcomes: the data extrapolated from the “revised studies” were collocated in tables, in particular, for each study specified, the author and year of publication and the study characteristics; (5) The analysis was carried out in the form of a narrative review of the reports. At the beginning of each section, the keywords considered and the kind of studies chosen have been reported. We evaluated, as suitable for the narrative review, the studies of any design that considered the relevance of diet, foods, nutrients, and dietary supplement for reducing the risk of loss of BMD.

## 3. Results

### 3.1. Overweight and Obesity

This research was carried out based on the keywords: “obesity” OR “overweight” AND “bone mineral density” OR “bone”. Fifteen articles were sourced: six prevalence studies, three observational studies, two cohort studies, one meta-analysis, one randomized clinical trial, one cross-sectional study, and one narrative review.

Appendix A shows the studies that evaluated the relationship between obesity/overweight and bone with their strength of evidence.

### 3.2. Physical Activity

This research was conducted based on the keywords: “physical activity” OR “physical exercise” AND “bone health” OR “bone mineral density” OR “osteoporosis”. Nine articles were sourced: one narrative review, two systematic reviews, three randomized controlled trials, one controlled prospective study, one prospective longitudinal study, and one clinical trial.

Appendix A shows the studies that evaluated the relationship between physical activity and bone with their strength of evidence.

### 3.3. Carbohydrates

This research was conducted based on the keywords: “carbohydrates” AND “osteopenia” OR “bone health”; twenty-one articles were sourced: one longitudinal study, one prospective cohort study, five cross-sectional studies, one randomized controlled trial, one randomized double-blind parallel study, six randomized double-blind cross-over studies, and six narrative reviews.

Appendix A shows the studies that evaluated the relationship between carbohydrates intake and bone with their strength of evidence.

Appendix A shows the studies that evaluated the relationship between carbohydrates supplementation and bone with their strength of evidence.

### 3.4. Lipids

This research was conducted based on the keywords: “osteopenia” AND “lipids” AND “bone health”; twelve articles were sourced: four cross-sectional studies, one randomized controlled trial, two clinical trials, four narrative reviews, and one systematic review and meta-analysis.

Appendix A shows the studies that evaluated the relationship between lipids intake and bone with their strength of evidence.

Appendix A shows the studies that evaluated the relationship between lipids supplementation and bone with their strength of evidence.

### 3.5. Proteins

This research was conducted based on the keywords: “protein intake” OR “protein supplementation” AND “bone” OR “osteoporosis”. Eighteen articles were sourced: five longitudinal studies, four systematic reviews and meta-analyses, four narrative reviews, two randomized double-blind placebo controlled trials, one cohort study, one randomized -cross-over trial, and one cross-sectional study.

Appendix A shows the studies that evaluated the relationship between proteins intake and bone with their strength of evidence.

Appendix A shows the studies that evaluated the relationship between proteins supplementation and bone with their strength of evidence.

### 3.6. Vitamin A

This research was conducted based on the keywords: (“vitamin A” OR “retinol”) AND “bone health” AND “humans”. Six studies were sourced: four narrative reviews, one double-blind crossover study, and one case-control study.

Appendix A shows the studies that evaluated the relationship between vitamin A intake and bone with their strength of evidence.

Appendix A shows the studies that evaluated the relationship between vitamin A supplementation and bone with their strength of evidence.

### 3.7. Vitamin D

This research was conducted on the keywords: “vitamin D” OR “vitamin D intake” OR “Vitamin D supplementation” AND “osteoporosis” OR “osteopenia” OR “bone mineral density”. Ten articles were sourced: one narrative review, one observational study, three meta-analyses and systematic reviews, two meta-analyses, and two clinical trials.

Appendix A shows the studies that evaluated the relationship between vitamin D intake and bone with their strength of evidence.

Appendix A shows the studies that evaluated the relationship between vitamin D supplementation and bone with their strength of evidence.

### 3.8. Vitamin E

This research was conducted based on the keywords: “alpha-tocopherol intake” OR “vitamin E intake” AND “bone mineral density” OR “fracture” OR “osteoporosis” AND “postmenopausal women”. Five articles were sourced: two systematic reviews, one meta-analysis, one cross-sectional study, and one Mendelian randomization study.

Appendix A shows the studies that evaluated the relationship between vitamin E intake and bone with their strength of evidence.

### 3.9. Vitamin K

This research was conducted based on the keywords: “vitamin K intake” OR “Vitamin K supplementation” AND “osteoporosis” OR “bone mineral density”. Ten articles were sourced: one meta-analysis, five narrative reviews, one observational study, and three randomized clinical trials.

Appendix A shows the studies that evaluated the relationship between vitamin K intake and bone with their strength of evidence.

Appendix A shows the studies that evaluated the relationship between vitamin K supplementation and bone with their strength of evidence.

### 3.10. Vitamins B

This research was conducted based on the keywords: “dietary B vitamins intake” OR “B vitamin supplementation” AND “bone” OR “osteoporosis”. Fifteen articles were sourced: two prospective cohort studies, five randomized, double-blind, placebo-controlled trials, three randomized controlled trials, one double-blind placebo-controlled trial, one randomized placebo-controlled trial, one case-control study, one longitudinal randomized trial, and one randomized double-blind trial.

Appendix A shows the studies that evaluated the relationship between vitamin B intake and bone with their strength of evidence.

Appendix A shows the studies that evaluated the relationship between vitamins B supplementation and bone with their strength of evidence.

### 3.11. Vitamin C

This research was conducted based on the keywords: “vitamin C intake” OR “vitamin C supplementation” AND “osteoporosis” OR “bone mineral density” OR “bone”. Seven articles and one database were sourced: three meta-analyses, two narrative reviews, one systematic review, one cohort study, and one integrated data system

Appendix A shows the studies that evaluated the relationship between vitamin C intake and bone with their strength of evidence.

Appendix A shows the studies that evaluated the relationship between vitamin C supplementation and bone with their strength of evidence.

### 3.12. Calcium

This research was conducted based on the keywords: “calcium intake” OR “bone health diets” OR “dairy products” AND “bone mineral density” OR “fracture risk”. Eight articles were sourced: one observational study, three systematic reviews, one systematic review and meta-analysis, and three meta-analyses.

Appendix A shows the studies that evaluated the relationship between calcium intake and bone with their strength of evidence.

Appendix A shows the studies that evaluated the relationship between calcium supplementation and bone with their strength of evidence.

### 3.13. Phosphorus

This research was conducted based on the keywords: “bone density” OR “bone disease” OR “osteoporosis” AND “phosphorus” OR “dietary phosphorus”. Nine articles were sourced: one observational study, two cohort studies, four narrative reviews, two systematic reviews and meta-analyses, and one set of guidelines.

Appendix A shows the studies that evaluated the relationship between phosphorus intake and bone with their strength of evidence.

Appendix A shows the studies that evaluated the relationship between phosphorus supplementation and bone with their strength of evidence.

### 3.14. Magnesium

This research was conducted based on the keywords: “magnesium intake” OR “magnesium supplementation” AND “bone” OR “osteoporosis”. Thirteen articles were sourced: four narrative reviews, two clinical trials, one retrospective cohort study, four observational retrospective studies, one cross-sectional study, and one systematic review.

Appendix A shows the studies that evaluated the relationship between magnesium intake and bone with their strength of evidence.

Appendix A shows the studies that evaluated the relationship between magnesium supplementation and bone with their strength of evidence.

### 3.15. Iron

This research was conducted based on the keywords: “iron intake” OR “iron supplementation” AND “bone” or “osteoporosis”. Five articles were sourced: two observational studies, two cross-sectional studies, and one randomized controlled trial.

Appendix A shows the studies that evaluated the relationship between iron intake ore serum levels and bone with their strength of evidence.

Appendix A shows the studies that evaluated the relationship between iron supplementation and bone with their strength of evidence.

### 3.16. Copper

This research was conducted based on the keywords: “copper” OR “Cu intakes” AND “bone health” OR “bone mineral density” OR “osteoporosis”. Nine articles were sourced: one systematic review (in vitro and animals), one meta-analysis, one observational study, one cross-sectional study, and five controlled prospective studies.

Appendix A shows the studies that evaluated the relationship between copper intake ore serum levels and bone with their strength of evidence.

Appendix A shows the studies that evaluated the relationship between copper supplementation and bone with their strength of evidence.

### 3.17. Zinc

This research was conducted on the keywords: “zinc” OR “zinc intake” OR “zinc supplementation” AND “osteoporosis” OR “osteopenia” OR “bone mineral density”. Six articles were sourced: one position paper, one cohort prospective study, one meta-analysis and systematic review, two cross-sectional studies and, one randomized double-blind placebo-controlled study.

Appendix A shows the studies that evaluated the relationship between zinc intake ore serum levels and bone with their strength of evidence.

Appendix A shows the studies that evaluated the relationship between zinc supplementation and bone with their strength of evidence.

### 3.18. Silicon

This research was conducted based on the keywords: “silicon intake” OR “silicon supplementation” AND “bone” OR “osteoporosis”. One article was sourced: one narrative review.

Appendix A shows the studies that evaluated the relationship between silicon supplementation and silicon intake and bone with their strength of evidence.

### 3.19. Manganese

This research was conducted based on the keywords: “bone density” OR “bone disease” OR “osteoporosis” AND “manganese” OR “oligoelements”; ten articles were sourced: three observational studies, two cohort studies, two narrative reviews, two randomized controlled trials, and one prospective cross-sectional study.

Appendix A shows the studies that evaluated the relationship between manganese intake ore serum levels and bone with their strength of evidence.

Appendix A shows the studies that evaluated the relationship between manganese supplementation and bone with their strength of evidence.

### 3.20. Boron

This research was conducted based on the keywords: “boron intake” OR “boron supplementation” AND “bone” OR “osteoporosis”. Two articles were sourced: one narrative review and one cross-sectional study.

Appendix A shows the studies that evaluated the relationship between boron intake and bone with their strength of evidence.

Appendix A shows the studies that evaluated the relationship between boron supplementation and bone with their strength of evidence.

### 3.21. Selenium

This research was conducted based on the keywords: “selenium intake” OR “selenium supplementation” AND “bone” OR “osteoporosis”. Four articles were sourced: one narrative review, one case-control study, one cross-sectional study and one cohort study.

Appendix A shows the studies that evaluated the relationship between selenium intake or serum levels and bone with their strength of evidence.

### 3.22. Water

This research was conducted on the keywords: “drinking water” OR “high-calcium mineral water” AND “hip fracture” OR “bone turnover” OR “postmenopausal women”. Four articles were sourced: one position paper, one set of guidelines, two observational studies, and one clinical trial.

Appendix A shows the studies that evaluated the relationship between water intake and bone with their strength of evidence.

Appendix A shows the studies that evaluated the relationship between water supplementation and bone with their strength of evidence.

### 3.23. Caffeine

This research was conducted based on keywords “bone” OR “osteoporosis” AND “caffeine”. Seven articles were sourced: two clinical studies, one cohort study, one meta-analysis, two systematic reviews, and one cross-sectional study.

Appendix A shows the studies that evaluated the relationship between caffeine intake and bone with their strength of evidence.

Appendix A shows the studies that evaluated the relationship between caffeine supplementation and bone with their strength of evidence.

### 3.24. Salt/Sodium

This research was conducted based on the keywords: “dietary sodium” OR “salt intake” AND “bone mineral density” OR “osteoporosis”. Six articles were sourced: one narrative review, one systematic review and meta-analysis, and four cross-sectional studies.

Appendix A shows the studies that evaluated the relationship between salt intake and bone with their strength of evidence.

### 3.25. Isoflavones

This research was conducted based on the keywords: “isoflavones” AND “bone mineral density” OR “osteoporosis”. Five articles were sourced: one narrative review, one randomized controlled trial, and three randomized double-blind placebo-controlled trials.

Appendix A shows the studies that evaluated the relationship between isoflavones supplementation and bone with their strength of evidence.

This review included 210 eligible studies, and the dedicated flowchart is shown in Figure 1, which represents graphically in a simple and intuitive way the proper nutrition for osteopenia/osteoporosis, specifying the quality and amount of food in order to reduce the risk of loss of BMD and to construct a food pyramid for osteopenic/osteoporotic subjects.

## 4. Discussion

### 4.1. Overweight and Obesity

Scientific literature has long studied the relationship between BMD and Body Mass Index (BMI). The study conducted by Veiga Silva et al., in 2015 analyzed a group of 1871 women with a mean age of 59.2 ± 10.9 years, mean weight of 68.7 ± 12.8 kg, and mean BMI of 27.7 ± 5.0 kg/m^2^ to evaluate the relationship between BMD, as assessed by Dual X-ray Absorptiometry (DXA) and BMI; the results of the work highlighted an inverse correlation between BMI and the risk of onset of osteopenia and osteoporosis and that, in particular, a BMI greater than 25 kg/m^2^ is a protective factor [13].

Another study carried out in 2017 evaluated the BMD, again by DXA, of 393 postmenopausal women with a mean age of 59.6 ± 8.2 years, reaching a conclusion similar to the previous one: the group of overweight women presented a BMD of the neck statistically significantly higher femur than the normal weight group. As regards the group of obese women, the association was even more evident [14].

Further confirmation comes from a more recent cross-sectional study conducted by Nayera E. Hassan in 2020 on a sample of 116 Egyptian women aged between 25 and 65 (average age of 48.85 ± 9.88). The women were divided into two groups: pre-menopausal women (51 subjects) and postmenopausal women (65 subjects), and BMD was measured by DXA, then compared between the various groups according to BMI. Once again, a positive correlation was established between not solely BMI and BMD, but also the amount of fat mass and its distribution also had a positive correlation with BMD; for example, the supra-iliac fold was positively correlated with the measured BMD at the level of the femoral neck [15].

Most of the prevalence studies analyzing the BMI-BMD ratio in the literature are often conducted on female populations, but the 2013 study by Salamat et al., investigated this correlation in a sample of 230 men (mean age 62.2 ± 8, 1 years), reaching the same conclusion: BMI > 25 kg/m^2^ is associated with a 2.2 lower risk of developing osteopenia and 4.4 times lower risk of developing osteoporosis [16].

Subsequently, again on the male population, a study was carried out in England in 2015 by Oldroyd and Dubey on 1263 male subjects; the work highlighted an increase in BMD in the lumbar and neck of the femur in subjects who had BMI values up to 35 kg/m^2^, while for higher BMI values, there would be no further benefits as regards BMD [17].

#### 4.1.1. Underweight

As for BMIs < 18.5 kg/m^2^, the literature mostly agrees that such low values of the BMIpredispose to an early reduction in bone mass.

For example, the 2016 Japanese study conducted by Tatsumi et al., investigated the relationship between underweight and osteopenia in a group of 749 Japanese women aged between 40 and 74 years by evaluating the relationship between BMD, measured through anthropometric and quantitative ultrasound, and BMI. The study concluded by confirming a correlation between underweight and an increased risk of developing osteopenic states [18].

The study by Lim and Park et al., published in the same year, also highlighted the presence of a correlation between underweight and low BMD, specifically in women. Of the 1767 women analyzed, among those with BMI < 18.5 kg/m^2^, the prevalence of BMD < −2 (at the lumbar level and at the level of the femoral neck) was found to be 23.9% in contrast to the prevalence of 9.4% found in normal-weight women (*p* < 0.001) [19].

Taking into consideration the sub-population of patients with anorexia nervosa (AN), often characterized by BMI < 17 kg/m^2^, the meta-analysis conducted by Hübel et al., (analyzing more than 60 studies concerning body composition in patients with AN) highlighted that the inadequate supply of nutrients reduces total bone mass; furthermore, the possibility of a full recovery of the same after normalization of the post-therapy BMI emerged [20].

#### 4.1.2. BMI and Body Composition

In 2006, Villareal et al., wanted to evaluate the effect of weight loss on the change in BMD by comparing three treatment groups: a group treated with the low-calorie diet only (CR group), a group treated with only physical exercise (EX group), and an untreated group who were given only behavioral advice for weight loss. The two treatment groups were made comparable since the weight loss was calculated on the same caloric deficit (−16%). The survey found that, in light of a very similar weight loss in the two groups (−8.2 ± 4.8 kg in the CR group, −6.7 ± 5.6 kg in the EX group), the loss of BMD was correlated with the weight loss in the group treated with caloric restriction (*p*-value = 0.007), an effect not observed in the exercise group that maintained lean mass [21].

Madeira et al., in 2014, further investigated the previous study by analyzing the correlation between the amount of fat mass and lean mass in relation to bone density in a population of obese subjects aged <50 years; the authors concluded that lean mass is related to BMD in obese subjects (whole femur BMD r = 0.385, *p* = 0.008; radio BMD r = 0.427, *p* = 0.003), while this correlation is not observable for fat tissue [22].

Similar conclusions also came from the recent observational study by Ji Lee et al. (2019), which wanted to evaluate the effect of obesity on bone density: the authors highlighted that a high BMI increases the density of the bones that support weight, while the rest are less subject to its effect; furthermore, the authors, further subdividing weight into lean mass and fat mass, concluded that in reality, lean mass is more related to BMD than fat mass [23].

#### 4.1.3. BMI and Fracture Risk

The BMI-fracture risk ratio (RR) is another correlation investigated in the literature for its “controversial” aspect: if, on the one hand, the benefit of increased weight can be quantified in greater bone density due to its “gravitational” effect of bone stimulation able to support the weight of the body, then, on the other, the problems inherent to hypomobility and functional impediment due to an excess of fat mass should theoretically predispose to fractures.

The 2013 cohort study by Tanaka et al., followed a group of 1614 Japanese menopausal women over time for an average of 6.7 years to assess fracture risk based on BMI: data showed that underweight women (RR = 0.45, *p*-value < 0.01) and normal weight (RR = 0.61, *p*-value < 0.01) are at lower risk of vertebral fractures than overweight/obese ones, and consequently, the subjects belonging to this last category are at greater risk of fractures of the spine [24].

Similar conclusions came from the narrative review conducted in Italy by Fassio et al., in 2018, elaborated by considering 36 studies on the subject present in the literature: the authors concluded that there is sufficient evidence regarding the fact that obesity tends to prevent certain types of fractures (e.g., hip fracture) while predisposing to others (e.g., lower limb fractures, excluding femoral neck fractures) [25].

The 2011 study by Nielson et al., focused on assessing the fracture risk in relation to BMI but in a population of only men aged > 65 years: in this case, it was possible to highlight that the subjects under examination who were overweight, obese I, and obese grade II (Hazard Ratio = 1.94) were at an increasing risk of developing non-vertebral fractures; however, the effect was no longer statistically significant after correcting the HR for the degree of movement limitation. This suggests that overweight/obese individuals are likely to be at greater risk of fractures due to their functional impediment to movement [26].

Finally, the observational study by Prieto-Alhambra et al., conducted on a large sample of female subjects aged >50 years (832,775 patients) from a vast database of data collected by general practitioners in Catalonia, always with the aim of evaluating the effect of BMI on fracture risk, showed that hip fractures are less frequent in overweight (RR = 0.77, *p* < 001) and obese (RR = 0.63, *p* < 0.001) than in those underweight or normal weight, while overweight and obese subjects are at greater risk of fracture of the proximal humerus (RR = 1.28, 0.018) compared to the latter [27].

Considering the 15 studies analyzed, including more than 120,000 subjects in total, it is possible to conclude that BMD is correlated with BMI: specifically, high BMIs tend to be associated with higher bone density for the same sex and age, probably due to the effect due to the greater weight exerted on the bone structure; low BMIs (underweight and severely underweight) are associated with a lower BMD, but not so much for the reduced weight exerted on the bone trabeculae as for nutritional deficiencies and concomitant state of hypo-nutrition. The effect of weight on bone density, however, is not related to the amount of fat mass present in the individual but rather to the amount of lean mass present in the subject, probably due to the greater use of the musculoskeletal system itself and therefore to greater stress of the bone system by the force of gravity. It is not possible to state that a state of overweight or obesity is in any case desirable to be able to maintain a good degree of bone mineralization also because many studies agree that these states predispose to a greater fracture risk. Therefore, it is possible to conclude that a good nutritional status, a BMI included in the normal range and overweight (BMI between 18 and 30 kg/m2), and a good degree of physical exercise can help to maintain a good bone mineralization index, while the state of obesity and that of underweight predispose to alterations that compromise good bone health.

Regarding the relationship between BMI and fracture risk, the results in the literature on this topic are fragmentary, but it is possible to conclude that variations in BMI outside the normal weight predispose to different fracture risks. This could be attributable to the fact that, at low BMI, one is more prone to traumatic or atraumatic fractures in the context of bone tissue tested by nutritional deficiencies, while at high BMI, the risk is more related to functional impediment due to the presence of fat itself.

### 4.2. Physical Activity

It is known that physical activity has numerous benefits on the general health of the individual. Physical activity is also considered an excellent support for bone health. In detail, as described in the narrative review by Troy et al. of 2018, during physical exercise, the forces transmitted through the skeleton on the bone generate mechanical signals that are recognized by osteocytes; they, in turn, trigger a cascade of biochemical responses that lead to an increase in bone turnover and net to bone deposition [28].

#### 4.2.1. Benefits of Physical Activity on the Bone of Growing Subjects

It is essential that, in adolescence, the highest possible peak of bone mass is reached in order to counteract the subsequent physiological decrease in bone mass. Furthermore, the higher the peak bone mass achieved during growth, the lower the risk of developing osteoporosis in the future [29]. It has been estimated that a 10% increase in peak bone mass would impart to additional 13 years of osteoporosis-free life for typical older woman [30].

As Troy et al., suggested in the 2018 review entitled, “Exercise early and often: effects of physical activity and exercise on women’s bone health,” the habit of practicing physical activity should be strongly encouraged already in the very young since, during the period of growth and development, adolescent bone is extremely sensitive to physical exercise and allows to increase the strength of the bone, which will tend to be better preserved with increasing age [28].

In a 2007 review by Hind et al., many randomized and non-randomized controlled studies were analyzed in order to define the effect of physical exercise on bone mineral growth in children and adolescents. Among the studies conducted in prepubertal age, the study with the strongest evidence, conducted by Fuchs et al. in 2001, demonstrated an increase in femoral neck bone mineral content (BMC), lumbar spine BMD, and lumbar spine BMC (measured with DXA) of 4.5%, 2%, and 3.1%, respectively, in exercisers (in particular, seven months of training lasting 10 min and performed three times a week based on jump circuits) compared with sedentary controls [31]. A similar study by Bradney et al. in 1998 observed similar results when prescribing an eight-month training program with activities such as aerobics, football, or volleyball [32]. Studies have thus shown a beneficial effect on bone mineralization in prepubertal subjects, both for medium- and high-impact activities (weight-bearing exercise). Furthermore, these studies have observed that the increase in size, density, and strength of the bone are qualities that persist over the years. Observing the subjects even in the seven months after the study period, Fuchs et al., recorded a 4% increase in femoral neck BMC [31]. The same was observed in a sample of growing girls subjected to nine months of training with high-impact jumps, followed by 20 months of normal activity, resulting in an increase in lumbar spine BMC of 6% compared to the control group [33]. Hind’s review concluded that although long-term effects are still unknown, maximizing peak bone mass is likely to offset future development of OS and bone fragility [34]. A prospective longitudinal study published by Baxter et al. in 2008 confirmed how, even after some time, subjects who practiced physical activity (understood as team sport, dance, or gymnastics) in childhood/adolescence maintain higher levels of BMC in young-adult age. The 154 participants recruited from 1991 to 1997 at the age of 8–15 (and grouped by physical activity level into “active” or “inactive”) returned for follow-up between 2002 and 2006 at the age of 23–30. Measuring BMC values in adolescence resulted in an 8% increase in BMC at the total-body level (TB), 13% at the lumbar level (LS), and 11% at the hip level among active males (total hip, TH) compared to the inactive ones, while in girls, there was an increase of 8% and 15% at the TB and LS level, always compared with the inactive group. Still measuring the same values in young adulthood, the group of active adolescents recorded an 8–10% increase in BMC, and that of girls was 9–10%. This demonstrates that practicing physical activity at a young age brings a benefit to bone health that will be maintained over the years [35]. To investigate whether the positive effect on the geometry of the cortical bone induced by physical activity carried out during growth persists in adulthood, even after such activity has ceased, Nilsson et al., conducted a study in 2009 on a sample of 1068 young male adults (aged between 18 and 20 years). Using standardized questionnaires, information on current and previous sports activity was collected in order to divide the sample into subgroups. A *p*-QTC device was used to scan the tibia and radius. The study showed that the group of inactive subjects, who had practiced sports while growing up, had better values of cross-sectional area, cortical thickness, and cortical periosteal circumference of the tibia than the group of subjects who had never practiced sports. In conclusion, therefore, Nilsson’s study reconfirms the beneficial role that sports activity practiced during growth has on the bone cortex and also demonstrates how this positive effect is maintained even after such activity has ceased [36].

#### 4.2.2. Physical Activity in Adulthood and Osteopenia

A 2018, single-blind randomized controlled study conducted by Watson aimed to evaluate the effect on BMD and evaluate any negative effects of a high-intensity resistance and impact training (HiRIT) and an unsupervised, low-intensity, home-based exercise (control group, CON) on osteopenic postmenopausal women. A total of 101 participants were recruited, with an age over 58 and a T-score < −1.0 in the lumbar or femoral area, and were then divided into two groups: one would follow the HiRIT-type training while the other the CON type. Each participant was evaluated for BMD (through DXA), anthropometric values, calcium intakes, and daily physical activity. For eight months, therefore, each individual followed a bi-weekly 30-minute training plan, performed at home or in a center, depending on which type of training she was assigned. At the end of the study period, the women belonging to the HiRIT group showed an improvement in bone strength indices in all the measured sites, even higher than those obtained by the CON group. Furthermore, since no type of adverse event (e.g., fractures) occurred during the study, it can be concluded that high-intensity training (HiRIT) is safe even in this population group despite low bone mass levels. This study shows how training in itself (especially at high intensity, resistance, and with weights) is conducive to maintaining good bone health, especially when performed adequately [37].

A more recent systematic review analyzed 18 observational studies on the relationship between physical activity and reduction of fracture events at various sites out of a total of 1,382,084 normal adult individuals with an age range ranging from 40 to 97 years. The study confirmed a positive correlation but emphasized that this is not determined solely by the positive effect that training has on the bone. In fact, practicing sports above all reduces the risk of falling, improving balance and coordination of movements. In subjects aged 60 or over, in order to maintain or improve bone mass, it would be recommended to combine resistance and strength exercises [38].

The International Osteoporosis Foundation (IOF) guidelines state that playing a sport, however, may prove difficult for people with low BMD due to fear of getting hurt, falling, and/or fracturing. In any case, however, it is concluded that the benefits of sports outweigh the risks since physical activity contributes to building and maintaining the strength of bones and muscles as well as improving the balance and coordination of movements [39].

In the conclusions of the review narrative by Troy et al., some recommendations were reported on the type of physical activity most suitable for different age groups. For example, in older women at risk of fracture, it is preferable to avoid high-impact sports, opting instead for weight-bearing activities, such as resistance training, yoga, or walking [28].

It can be concluded by saying that physical activity is certainly useful in reducing the risk of fracture both directly, leading to an increase in BMD in certain sites and, indirectly, reducing the risk of falling.

However, the evidence remains that it has a wider preventive role when carried out both at a younger age and when carried out regularly throughout the life span [35,36]. Given its wide diffusion, its low cost, and its adaptability, it is very important that physical activity is therefore encouraged already in the very young so as to make it a habit and ensure that it is practiced for life, carrying out its preventive role also against osteopenia and osteoporosis.

#### 4.2.3. Recommendations on the Type of Physical Activity

In the review published in 2019 by Cauley et al., various guidelines and recommendations published by various scientific societies have been grouped and summarized. All agree that the best option in the patient already suffering from osteopenia or osteoporosis is a combined training program. For example, Canadian guidelines (“Osteoporosis Canada: Too Fit to Fracture”) recommend two or more times a week progressive resistance training, daily balance exercises, and 150 min a week of aerobic physical activity [38].

The same is true of the guidelines for the diagnosis and treatment of postmenopausal osteoporosis updated to 2020; that is, to reduce the risk of falling and to strengthen the muscles in this category of subjects, it is advisable to combine weight training with resistance training [40].

The IOF guidelines also give indications on the type of training recommended according to the age group. In particular, these guidelines recommend for all osteoporotic patients to carry out a training program focused on posture, balance, gait, coordination, and hip and trunk stabilization rather than general aerobic fitness. This training program should be supervised by a doctor or physiotherapist and designed on the specific needs and abilities of the subject [39].

In adults who want to prevent bone loss and maintain muscle strength, the goal should be to perform three or four times a week for 30 to 40 min certain activities, such as weight-bearing exercise (dancing, aerobics, jogging, walking, and playing sports are all excellent, but exercises with less impact, such as using a treadmill, elliptical strider, or step machine at home or in the gym, are also a great way to strengthen bones), muscle strengthening/resistance exercise, and balance exercise (for example tai chi, yoga or Pilates) [39].

With regard to children and adolescents, on the other hand, to increase bone mass and density and protect the skeleton from the onset of osteoporosis in the future, it is recommended to carry out high-intensity and short-duration workouts and at least 40 min a day of physical activity. The latter should include sports with an element of overload along with activities, such as dancing, skiing, running, jumping, or walking [39].

For all osteoporotic patients. The guidelines recommend conducting a training program that focuses on posture, balance, gait, coordination, and hip and trunk stabilization rather than general aerobic fitness. This training program should be supervised by a doctor or physiotherapist and designed on the specific needs and abilities of the subject [39].

### 4.3. Carbohydrates

A study was recently published that deals with the relationship between carbohydrate-containing food intake and bone. In 2019, Matsuzaki and colleagues evaluated the effects of brown rice intake on BMD over a period of one year; 40 subjects with an average age of 73 years were enrolled and divided into two groups for whom the bone area was assessed with the calcaneus quantitative ultrasound (QUS) at the beginning and at the end of the study: the first group consumed 100 g of white rice and 100 g of brown rice per day, while the second group consumed 200 g of white rice per day. The results of the study reported a significant change in bone area in the group that ate brown rice compared to the group that ate white rice, with a statistically significant increase in this area after 12 months of treatment in the group that ate brown rice and a decrease in the group that ate white rice [41]. There are no other studies in the literature that consider the assessment of BMD in association with the consumption of carbohydrate-based foods, such as rice, pasta, or bread and their derivatives.

In addition to starch and sugars, dietary fibers are also part of the food group of carbohydrates, among which we also find prebiotics, i.e., molecules that are not used directly by the human body for energy purposes but instead become substrates for the intestinal microbiota, such as oligosaccharides. These are represented by fructo-oligosaccharides (FOS) (which includes inulin), beta-glucans, galacto-oligosaccharides, gluco-oligosaccharides, soy-oligosaccharides, and isomalt-oligosaccharides (IMO). Several searches in the literature have analyzed their association with bone health.

In a study of 2018, the results of the Framingham Offspring Study were reported on the association between fiber consumption and BMD; 792 men and 1065 women were enrolled and asked to fill out a questionnaire on food frequencies and subjected to BMD analysis via DXA carried out at the beginning of the study, after four years, and again after eight years. The average fiber consumption was comparable between men and women (19.7 g in men and 19.5 g in women) and was subsequently divided into quartiles based on daily fiber consumption. The results showed a protection in the loss of bone mass at the level of the femoral neck in men belonging to the upper three quartiles of fiber intake compared to the lowest quartile; no significant associations emerged in women [42].

In 2019, the study by Lee T et al., enrolled 2187 subjects undergoing 24-hour recall to assess fiber consumption associated with BMD at different sites as assessed by DXA. The subjects were divided by gender and age groups (18–45 years, 46–65 years, and >65 years). Average daily fiber intake values were 9.08 g for men and 6.34 g for women. In women, fiber intake was not positively correlated with BMD values at any site and for any age group, while in men, there was a positive correlation between fiber intake and BMD in L1 and L2 in the age group between 18 and 45 years (average fiber intake 8.47 g); no significant correlations were found in the other age groups of men and in the other sites investigated [43].

A subsequent study in 2021 took into consideration the quantity of fiber assumed (investigated through a frequency questionnaire) and the BMD values evaluated with calcaneal ultrasound in 384,134 subjects; the average consumption of fiber was slightly higher in women than in men (14.5 g/day against 14 g/day), while average values of heel bone density were found to be higher in men than in women. After adjusting for age, gender, and BMI, the results showed a significant association between the consumption of fiber (cereals, fresh and dried fruit, cooked and raw vegetables) and heel BMD. Sensitivity analyses were also conducted in a subgroup of subjects by evaluating the association between fiber intake and BMD assessed by DXA: the results showed that a higher fiber intake was correlated with a better total body BMD and a lower risk of hip fractures when adjusted for age, gender, and BMI. However, these associations were not significant when further adjusted for physical activity, smoking, alcohol intake, dietary calcium and vitamin D intake, and total energy intake. Subsequently, the study wanted to examine whether genetic variations in polygenic scores (PGS) linked to the production of microbiota-derived short-chain fatty acids (SCFA) could modify the association between dietary fiber intake and heel BMD. A significant interaction was therefore found between the PGS of propionate and dietary fiber intake on bone health. This association was then found to be stronger among participants with lower genetically determined fecal propionate. However, no interaction was found with the PGS of butyrate. Therefore, it is possible to hypothesize that higher levels of propionate (genetically determined and derived from the gut microbiota) could attenuate the association between dietary fiber and BMD. In this case, higher dietary fiber intake may therefore have a less beneficial effect on bone health. It was therefore possible to conclude that higher intakes of dietary fiber (total and subtypes) from various food sources are associated with a higher BMD of the heel. Additionally, participants with lower genetically determined propionate production may benefit more from consuming more dietary fiber [44].

The correlation between fiber intake and BMD could be meditated by the composition of the microbiota: a recent narrative review summarized the findings from preclinical studies that support that gut microbes positively impact bone mineral density and strength parameters [45]. Moreover, an even more recent narrative review reported that both the in-vitro and in-vivo studies show that the probiotics that positively affect bone health are Lactobacillus and Bifidobacterium through calcium regulation either on the transcellular pathway or on the paracellular pathway [46].

Other studies have evaluated the specific association with fruit and vegetables. In a 2006 cross-sectional study, the association between fruit and vegetable consumption and BMD was evaluated in a group of postmenopausal women aged between 48 and 63 years; subjects completed a food frequency questionnaire and performed an assessment of lumbar and femoral BMD using DXA. The results showed an average intake of vegetables equal to 295 g per day and fruit equal to 175 g per day and a favorable independent association between the consumption of fruit and vegetables and BMD at the level of the total body, lumbar spine, and hip was demonstrated even after making adjustments for age, weight, height, years of menopause, total energy intake, and protein and calcium intake. In addition, the subjects were stratified into quintiles on the basis of fruit and vegetable intakes, and the average BMD values in the sites listed above increased by 2.1%, 4.1%, and 2.2%, respectively, more in the subjects in the major quintile than in the subjects present in the bottom quintile of fruit and vegetable intake. The results therefore report an independent positive association between fruit and vegetable consumption and BMD at all sites; the hypotheses proposed by the authors are two: the first is the acid-base theory, where it is postulated that the acid load (mainly due to the consumption of protein foods, phosphorus, and chlorine) is partly buffered by the bone mineral; the second hypothesis is related to the intake of some vitamins (such as vitamin C and vitamin K) and phytoestrogens, which are very present in fruit and vegetables [47].

In 2012, with the same working methodology of the study previously described (food frequency questionnaire and DXA), a cross-sectional study was carried out with a sample of 816 subjects divided into male and female—boys, young women, and postmenopausal women—and subsequently divided in tertiles based on the amount of fruit and vegetables eaten; the results of the study report a positive association in all subjects with greater benefits found in male boys and postmenopausal women, with stronger associations linked to fruit consumption than vegetable consumption; furthermore, improvements in BMD and BMC are more evident in the tertile of higher consumption than in the lower one as regards fruit intake, while as regards vegetable intake and overall fruit and vegetable intake, the difference between tertile upper and lower was modest. The hypotheses proposed are the same as reported in the previous study even if, for the authors, these theories do not fully explain the better results associated with the consumption of fruit compared to vegetables; for this reason, the proposed explanation is linked to the consumption of salt/sodium as a condiment used for vegetables, which would tend to counteract its positive effects [48].

More recently, in 2017, Qiu et al., considered 3089 subjects (2083 women and 1006 men) aged between 40 and 75 who underwent DXA and a frequency questionnaire. The subjects were divided into sex-specific tertiles based on fruit and vegetable intake. The results show a dose-dependent association between the total intake of fruit and vegetables and BMD and the risk of osteoporosis. Mean BMD values were higher in the highest tertile than the lowest in total fruit and vegetable intake in the femoral neck and hip sites; with regard to the intake of fruit only, the average BMD values were higher in the highest tertile than in the lowest tertile in the whole body, hip, and femoral neck sites, while no significant associations were found with vegetables alone. Additionally, in this case, the authors took into consideration all the hypotheses found in the other works (acid/base hypothesis, positive effect of vitamins C and K, effect of phytoestrogens, and negative effect of sodium/salt). As regards the onset of osteoporosis, the favorable association with the consumption of fruit and vegetables was evident only in subjects with a BMI of less than 24 kg/m^2^. The explanation provided by the authors is linked to the fact that subjects with lower BMI may have a greater bone oxidative stress, and consequently, the positive effects of the antioxidants contained in fruits and vegetables are shown more in contrast to the development of osteoporosis; moreover, individuals with higher BMI had better bone mass giving less space for antioxidants contained in fruits and vegetables to demonstrate their benefits [49].

Consumption of dried fruit was also evaluated in correlation with BMD. In 2011, in a randomized controlled trial in reducing the risk of osteoporosis evaluated 236 postmenopausal women, of which 100 completed the one-year study; the subjects were divided into two groups, with an intake of 100 g of dried plums or 75 g of apples. BMD was assessed through DXA at the beginning and end of the study. In both cases, a positive change in BMD was reported but with a greater positive effect in the group that took dried plums and more pronounced at the level of the ulna and spine. In addition, with regard to bone turnover markers, only with the intake of dried plums was there an improvement in bone-formation indicators [50].

To evaluate a possible dose-dependent effect, the same authors in 2016 enrolled 48 osteopenic postmenopausal women and divided them into three treatment groups for six months: 50 g of dried plums (group 1), 100 g of dried plums (group 2), and control group; the assessment of bone densitometry was carried out right at the beginning and at the end of the study. Both supplemented quantities proved useful in preventing the reduction of total bone mass compared to the placebo group; analyzing the specific sites, there were no statistically significant differences between the groups for BMD at the hip and ulna level, while an increase trend was indicated at the level of the spine in patients treated with both 50 g and 100 g compared to untreated group although this variation was not significant. Bone-resorption markers were reduced already after three months in subjects who took dried prunes, with no significant differences between those who had taken 50 g or 100 g, demonstrating that even small doses of dried prunes can be useful in the prevention of reduction of bone mass in postmenopausal osteopenic women [51].

Confirming what has been reported so far in this paragraph, a 2018 review evaluated the relationship between fiber intake and various diseases, including bone loss. In particular, in the review of the literature, it was highlighted that an adequate intake of whole fruit and vegetables (at least 0.5 kg per day) was associated with a better BMD, especially during adolescence, with a slowing down of the reabsorption processes of bone [52].

#### 4.3.1. Supplementation with Prebiotics

Prebiotics represent a particular type of fiber useful for the growth and maintenance of the intestinal bacterial flora; some studies have evaluated their relationship with bone metabolism.

As early as 1999 and later in 2000, Van den Heuvel and colleagues assessed calcium absorption in postmenopausal women after taking lactulose or trans-galacto- oligosaccharides (HRT), respectively, in two randomized double-blind cross-over studies; in both studies, all subjects took both supplementation and placebo with 19 days of wash out between treatments; in the study of 1999, 12 subjects were enrolled and divided into three groups, with an average age of 60.5 years, who were given a supplement of 5 or 10 g of lactulose powder or placebo for nine days; at the end of this period, the 19 days of wash out began, and subsequently, the subjects took the remaining two treatments. To conclude, an increase in calcium absorption was observed after lactulose consumption, depending on the dose taken compared to placebo treatment and not accompanied by an increase in urinary calcium excretion [53]. Similarly, in 2000, 12 subjects were enrolled and divided into two groups, with a mean age of 62 years, who were given a supplement of 20g of HRT (gradual increase, starting from 10 g on day one) or placebo for nine days, with 19 days of wash out between treatments. Again, as in the previous similar study, an increase in calcium absorption was observed after consumption of the HRT-rich product compared to placebo treatment and was not accompanied by an increase in urinary calcium excretion [54].

In a population sample that included 26 postmenopausal women, an intake of 8 g for three months was evaluated per day of chicory fructan fiber: the data collected showed a decrease in serum levels of bone-resorption markers, meaning a slowdown in bone degradation processes [55].

In a 2014 paper, an integration was carried out with 12 g of soluble corn fiber for three weeks in a group of 24 adolescents with a low calcium diet aged between 12 and 15 years: through the use of double stable isotope tracers, a stimulation of calcium absorption was found in the intestine by the microbiota; [56] on the basis of this study, we wanted to verify the effects of soluble corn fiber on 14 postmenopausal women at risk of osteoporosis; each of them took 0, 10, or 20 g per day for 50 days in random order, with a consequent finding of improvement in bone-formation markers proportional to the amount of fiber assumed [57]. These considerations were later confirmed in a 2018 review, which stated that prebiotics could prove to be a nutritional prevention strategy against osteoporosis since an improvement in calcium absorption can counteract, at least in part, the problem of a reduced intake of this mineral. The review concludes by recommending to place a greater focus on those foods that naturally contain prebiotics and to incorporate them within the dietary recommendations for bone health [58].

#### 4.3.2. Simple Carbohydrates

Even the intake of simple sugars in the diet would seem to affect bone health. A 2015 study aimed to investigate the possible correlation between serum glucose levels and bone health. Four thousand forty-eight postmenopausal Chinese women were involved and divided into two groups based on the results of the QUS assessment: osteopenic group and non-osteopenic group. Anthropometric data and a blood sample were collected from each participant, and they were subjected to a questionnaire on lifestyle and eating habits; subsequently, a glucose tolerance test was conducted with the intake of 75 g of glucose. The study showed a relationship between systolic blood pressure (SBP), fasting blood glucose (FBG), postprandial blood glucose (PBG), glycated hemoglobin (HbA1C), and osteopenia in postmenopausal women, while no significant relationship was observed between dyslipidemia, diastolic blood pressure (DBP), and osteopenia even after controlling for multiple confounding factors. The author of the research stated that the evolution processes of bone tissue in patients at risk of diabetes could be complex, and therefore, further studies are needed [59]. Diabetes also seems to cause damage to bone tissue: the latter, due to its extension, requires a high amount of energy for its renewal processes. This energy is therefore obtained thanks to the glucose present in the blood, the use of which is determined by the levels of insulin. In diabetic patients, a different pathological mechanism has been hypothesized: a reduced process of bone renewal could be due to the inability to use the glucose present in the blood due to a lack of adequate insulin levels, consequently causing microfractures in the tissue itself [60]. In fact, in diabetic patients, the reduction of bone strength is associated with an increase in cortical porosity that is not accompanied by a loss of trabecular bone, as occurs in patients with senile or menopausal osteoporosis. A 2018, review analyzed the relationship between diabetic pathology and bone, confirming that type 2 diabetes is associated with an increased risk of fracture, especially at the hip level [61].

In conclusion, it can be said that an adequate intake of fiber, belonging to the carbohydrate group, represents an important source of nutrition for the bones; optimal intakes of fruit (especially) and vegetables have been shown to be positively correlated with better BMD. Furthermore, prebiotics, used with the aim of influencing the composition of the human microbiota, prove to be capable of significantly influencing the state of bone tissue throughout life, thus helping to reach an adequate bone peak and subsequently to slow down the process of bone loss, further intervening in a positive way in the processes of calcium absorption in the intestine.

An adequate consumption of fruit and vegetables with a good intake of fiber is therefore recommended equal to three portions of fruit (approximately 150–200 g per portion) and two portions of vegetables (approximately 200 g per portion) per day. Among the various types of fruit, the richest in fiber are currants (7.8 g/100 g), raspberries (7.4 g/100g), pear (3.8 g/100 g), blackberries (3.2 g/100 g), and kiwifruit (2.2 g/100 g); among the various types of vegetables, those more rich in fiber are: mushrooms (6.5 g/100 g), artichokes (5.56 g/100 g), brussels sprouts (5.2 g/100 g), chicory (3.6 g/100 g), and broccoli (3.1 g/100 g). It is also desirable to consume more foods that contain prebiotics, such as chicory, artichokes, onions, leeks, asparagus, garlic, beans, bananas, and corn, which contain inulin; barley, oats, and mushrooms, which contain beta-glucans; honey, containing isomalt-oligosaccharides; and soy, which contains soy-oligosaccharides. The intake of simple sugars, on the other hand, should be avoided.

### 4.4. Lipids

The scientific literature has investigated the association between BMD and lipid intake. In particular, omega 3 and olive oil have been investigated. Three studies investigated the association between BMD and omega-3 intake as assessed by questionnaires on the frequency of food consumption.

In 2007, the Zalloua study evaluated how eating habits can affect BMD, involving 12,055 subjects divided into four groups: men with, respectively, age ≤45 and >45 years were part of group 1 and 2, while in groups 3 and 4, premenopausal and postmenopausal women were included, respectively. BMD was stratified into quartiles, using only the lower and upper quartiles in the analysis. For premenopausal women (group 3), subjects with higher fish consumption (>250 g per week) in the lower BMD quartile were 25.5% compared to 34.4% in the upper BMD quartile, while subjects with a consumption of lower fish (≤250 g) in the lower quartile were 74.5% compared to 65.6% in the upper quartile of total body BMD. The increase in fish consumption was positively associated with a better BMD (*p* < 0.001). The results therefore support the idea that fish consumption has a positive effect on BMD [62].

A 2017 cross-sectional study aimed to study the effect of omega-3s on a young female population. The study group collected 275 participants with an average age of 20.6 ± 1.4 years and an average BMI of 21.2 ± 2.7 kg/m^2^. Each participant was measured the BMD in the hip and lumbar area using DXA, and the intakes of calcium, vitamin K, vitamin D, phosphates, and unsaturated fatty acid(PUFA) were obtained through a food frequency questionnaire. The study results show that eating foods containing omega-3 fatty acids significantly contributed to total hip BMD but not lumbar BMD [63].

Subsequently, Lavado-Garcia and colleagues studied the effectiveness of omega-3 intake on BMD in Spanish middle-aged women. The study gathered 1865 participants, with an average age of 54 ± 10 years and an average BMI of 27.3 ± 4.5 kg/m^2^. Each subject underwent DXA to measure lumbar and hip BMD and completed a food frequency questionnaire. Of the 1865 participants, 194 suffered from osteoporosis and 707 from osteopenia, while 964 showed normal T-score values. In the groups of normal and osteopenic women, omega-3 intake ((eicosapentaenoic acid) EPA + docosahexaenoic acid (DHA) 0.52 ± 0.21 g/day) was positively associated with better BMD at both sites, while the same could not be said for the group of osteoporotic participants, in which no association emerged [64].

In 2009, Maggio and colleagues evaluated the impact of omega-3s on osteoporosis in a review: nine studies carried out on humans were considered for a total of 17,720 individuals both in terms of intake of omega 3 with nutrition and of integration; the conclusions of the review stated that it is not possible to determine the therapeutic value of these compounds in clinical practice, and further future studies on the relationship between omega-6 and omega-3 are needed [65].

A 2013 review emphasized the role of inflammation as a cause of osteoporosis: the authors proposed the hypothesis of how a high intake of n-6 PUFA combined with a low intake of n-3 PUFA promotes a state of inflammation and low-grade chronic disease and a consequent reduction in osteoblastic factors, which thus contribute to osteoporosis [66].

A recent narrative review wanted to analyze the possible therapeutic role of PUFAs in the treatment of some pathologies involving the bone. Among the works cited, nine concerned osteoporosis and the risk of fractures: it was possible to conclude that PUFAs mainly exert protective functions on the bone by promoting the functions of osteoblasts and inhibiting the activities of osteoclasts [67]. In addition to the intake of omega-3s with food, their integration was also evaluated.

A 1995 paper by van Papendorp colleagues evaluated the effects of fish oil supplementation on markers indicative of bone metabolism; 40 elderly women were enrolled, with an average age of 80 years and with a previous diagnosis of osteoporosis made through DXA and who were divided into four groups that took for 16 weeks: 4 g of evening primrose oil, 4 g of fish oil (consisting of 16% EPA and 11% DHA), 4 g of a mixture of evening primrose oil and fish oil, and 4 g of olive oil. This short-term study suggested that supplementation with fish oil or in particular a mixture of fish oil and primrose oil can improve bone formation, as indicated by the increase in osteocalcin, the increase in procollagen, and the reduction of alkaline phosphatase [68].

In 1998, Kruger et al., investigated the effects of omega-3 supplementation on BMD in elderly women: 65 women with an average age of 79.5 years, with osteopenia or osteoporosis and low calcium intake, were involved and divided into two groups. One group received supplementation of 600 mg of calcium and 6 g of LCPUFA-rich oil, while the control group received coconut oil supplementation as a placebo in addition to calcium. After 18 months, the control group experienced a 3.2% reduction in lumbar BMD, while the group that received LCPUFA-rich oil supplementation did not change [69].

It is therefore clear the importance of not only controlling the amount of fats to be taken but also their quality in trying to take in adequate doses those fats, such as omega-3s, which prove to be useful allies in contrasting the loss of bone mass and in the prevention of pathological states associated with these processes.

The most commonly consumed foods that contain omerga-3 belong both to the animal world and to the plant world, precisely blue fish (such as sardines, anchovies, mackerel), salmon, seaweed, nuts, oil, and flax seeds and chia seeds.

Considering extra virgin Olive (EVO) oil, this food has always been considered an extremely important food in the Mediterranean diet: there are numerous benefits associated with its intake, chiefly the positive effects it has on cardiovascular health. However, in more recent times, positive effects have also been discovered on bone health.

A systematic review in 2014 considered 37 articles on the relationship between bone health and the Mediterranean diet with particular attention to the intake of olive oil. The papers reported in the review, which concern both in-vitro studies and those in the animal model and in humans, underline the benefits of the Mediterranean diet in terms of reducing the loss of BMD and reducing the risk of fracture. In addition, the specific role of olive oil was evaluated. In the animal model, it was shown that an integration of olive oil led to an increase in BMD, an increase in plasma levels of alkaline phosphatase, and a reduction in bone resorption. The work carried out in humans (No. 3) has also shown that the intake of olive oil is positively associated with BMD with a reduction in the risk of fractures compared to the population of the United States, thus far from the model of the Mediterranean diet. The components potentially responsible for these benefits are phenols. Among these phenols, oleuropein, tyrosol and hydroxytyrosol, and lutein stand out, for which efficacy was evaluated in vitro (four papers) and in vivo on animals (four papers). Oleuropein increases the formation of osteoblasts of bone marrow stem cells, while in ovariectomy rats, it reduces the loss of bone mass due to the modulation of inflammation; tyrosol and hydroxytyrosol at the cellular level both inhibited the formation of osteoclasts in a dose-dependent manner and stimulated the deposition of calcium ions in the extracellular matrix, while at the animal level, a suppression of trabecular bone loss of the femur and prevention of bone loss caused by inflammation was observed; lutein reduced the differentiation of mononuclear cells into osteoclasts, while, when administered in ovariectomized mice, it favored a significant increase in BMD compared to controls. Therefore, the positive action of phenols on bone health seems to be linked to the reduction of inflammatory process [70].

Two other works followed this review. A 2018 observational cohort study investigated the effect of olive oil consumption on osteoporosis-related fracture risk in a middle-aged and elderly Mediterranean population. Eight hundred and seventy participants aged between 55 and 80 years with high cardiovascular risk were included and randomly divided into three groups: Mediterranean diet with extra virgin olive oil (greater than or equal to 50 g per day), Mediterranean diet with walnuts (30 g per day), or a low-fat diet. To evaluate the consumption of the various types of oil, a validated questionnaire on the frequency of food intake was used, administered at the start of the study and repeated every year during the follow-up period (8–9 years). Information on osteoporotic fractures totals were obtained from a systematic review of medical records. Four hundred fourteen osteoporosis-related fracture cases were documented during follow-up. There were no significant differences between the groups on fracture risk. Subsequently, the group that took extra virgin olive oil was divided into tertiles on the basis of which the common olive oil (refined and pomace oil) was also evaluated, adding it to the extra virgin olive oil to measure the total oil consumption. It was therefore shown that the participants belonging to the highest tertile of extra virgin olive oil consumption (average total oil consumption: 56.5 g/day) had a 51% lower risk of fracture compared to those in the lowest tertile. Therefore, the consumption of extra virgin olive oil (on average greater than 50 g per day) reduced the risk of fracture [71].

A further clinical study from 2018 evaluated the association between olive oil intake and the microarchitecture of cortical and trabecular bone in Spanish women. The study analyzed 523 women between the ages of 23 and 81. Participants underwent DXA and peripheral quantitative computed tomography, while the dietary intake of olive oil was assessed through a food frequency questionnaire; the median consumption of olive oil in the sample was 18.32 g/day, and the study subjects were divided into two groups: consumption ≤ 18.32 g/day (294 subjects) and consumption >18.32 g/day (229 subjects). The study observed positive associations, after adjustments, between total, trabecular, and cortical BMD and higher olive oil consumption (>18.32 g/day) in Spanish women [72].

In conclusion, the studies agree in highlighting that an adequate consumption of healthy fats (such as those contained in fish and extra virgin olive oil) is correlated with a reduced risk of bone loss; the reason is secondary to the anti-inflammatory activity carried out by the omega 3 fatty acids contained in the fish and by the polyphenols contained in the EVO oil. In particular, studies suggest to take at least 20 g per day of extra virgin olive oil (the ideal is about 50 g per day) and to take four portions a week of fish, particularly rich in omega 3 (intake of EPA + DHA at least 0.52 g/day), such as blue fish.

### 4.5. Proteins

In past years, it has often been assumed that excessive protein intake could be harmful to bone tissue, as indicated by several authors who have documented an increase in urinary calcium following increased protein intake [73,74], thus hypothesizing that an excess of animal protein intake in the diet could increase the production of endogenous acid, thus causing acidosis and the loss of calcium in the urine to a greater extent than proteins from plant sources [75,76], and that their excessive intake could be related to a higher incidence of forearm fractures [77].

However, all these studies have limitations, as the subjects analyzed were mainly young and healthy, often with a very low numerical sample. Furthermore, considering the different methods of analysis, it was often not possible to estimate the absorption of calcium in the intestine, and thus it was also not possible to fully examine the effect of proteins on calcium excreted in the urine.

In recent years, therefore, there is always increasing evidence that the increase in protein intake exceeding the recommended value of 0.8–0.9 g/kg/day, in some moments of life, such as old age, can bring important benefits from the point of view of bone health thanks to an improvement in bone turnover markers and BMD and a lower fracture risk, as evidenced by a recent narrative review of the literature [78].

Furthermore, some studies have shown that an inadequate protein intake can be negative on bone health. In a 1997 study, it was shown that a low protein diet (0.7 g/kg/day) but balanced in all other nutrients, including calcium, carried out in 14 healthy young women led to secondary hyperparathyroidism, with serum parathyroid hormone (PTH) levels increased 1.5–2.4-fold by day four and 1.6–2.7-fold by day 14 compared to values observed in subjects consuming moderate protein intake (1.0 g/kg/day) [79]. The data were confirmed by the same author in a subsequent study in 2000, highlighting increased levels of PTH after four days of a low-protein diet (0.7 and 0.8 g/kg/day), while no alterations were recorded in the case of a diet with protein content equal to 0.9 and 1 g/kg/day [80]. This same result was also reached in 1999 by a study by Giannini et al., conducted on 18 patients (10 men and 8 women aged 45.6 ± 12.3 years) with idiopathic hypercalciuria and kidney stones. A diet was administered containing 0.8 g of protein/kg/day and 24 mmol of calcium: urinary calcium excretion decreased, and serum PTH increased within 15 days of protein intake restriction [81].

A 2008 narrative review showed how, especially in postmenopausal women and men aged >65 years (about 1000 subjects), diets rich in proteins (up to 2.1 g/kg/day) are associated with an improved bone mass, mainly assessed by urinary calcium excretion and Double Photon Beam Absorbimetry (DPA), and fewer fractures, as long as an adequate calcium intake is guaranteed [82]. In a subsequent narrative review in 2014 investigating the relationship between protein intake, BMD and fracture risk in 21,556 subjects, mostly postmenopausal women and men aged >65 years, it was found that greater protein intake (greater than 0.8 g/kg/day), especially of animal origin, may be beneficial for bone health in conditions of adequate calcium intake [83]. A 2009 systematic review with meta-analysis analyzed 61 studies (31 cross-sectional, ecological, and cohort studies; 19 studies with supplementation that examined BMD, bone mineral content (BMC); and 11 studies of cohort and case-control that examined the risk of fracture) with a total of 210,540 subjects (peri- and postmenopausal women and men 30–89 years). It has been shown that there is a small positive effect of proteins (0.6–1.5 g/kg/day) on bone health, assessed mainly by DPA, single photon absorptiometry (SPA), DXA, QUS, and bone turnover marks, and the benefit may not necessarily translate into a reduced long-term fracture risk [84].

A further 2017 meta-analysis systematic review, in which 36 studies (16 randomized clinical trials and 10 cohort studies) were analyzed with a total of 454,002 subjects, mainly postmenopausal women and healthy adult men, showed that the effect of dietary proteins (15–30% of energy requirements, of both animal and vegetable origin) on the skeleton is to a small extent favorable without however showing any significant relationship between dietary proteins and fracture risk [85].

Further evidence supporting higher protein intake for better bone health is provided by a recent systematic review and meta-analysis by Groenendijk et al. The main aim of this study was to investigate the impact of a dietary protein intake greater than 0.8 g/kg/day (from any source) on BMD, BMC, bone turnover markers, and fracture risk in the elderly compared to a lower intake of protein in the diet. Therefore, 12 cohort studies and one randomized clinical trial were analyzed for a total of 273,087 subjects (mainly postmenopausal women and men with an age >65 years): the qualitative evaluation could show a positive trend among higher protein intakes and higher BMDs of the femoral neck and total hip. The meta-analysis of four of the cohort studies also showed that higher protein intake can lead to a significant decrease in hip fractures (−11%). However, it was not possible to draw conclusions regarding BMC and bone turnover markers [86].

A systematic review with meta-analysis from 2017 also came to the same conclusion, including 16 randomized clinical trials (1401 subjects, men and women pre-post menopause) and 13 cohort studies (271,963 subjects, men >50 years and postmenopausal women), with the aim of examining the relationship between protein intakes equal to or greater than 0.8 g/kg/day from any source and fracture risk, BMD, BMC, and bone turnover markers. Specifically, the meta-analysis of prospective cohort studies showed how high versus low protein intakes can lead to a statistically significant decrease in hip fractures (−16%). Furthermore, the data from the studies included in these analyzes lean towards the hypothesis that protein intake above the current recommended level is beneficial for BMD at several sites [87].

All these analyses are confirmed by the fact that osteoporosis, with increasing age, can often be related to the loss of muscle mass and therefore to sarcopenia. In this regard, Lima et al. [88] found that osteoporosis has an incidence of 19.2% in pre-sarcopenia but can increase up to 35.3% in sarcopenia. Therefore, the loss of muscle strength and muscle mass due to aging increases the risk of both sarcopenia and osteoporosis [89].

In fact, it should be remembered that to maintain and recover the muscles, elderly subjects need a greater protein intake than younger subjects; elderly people should therefore have an average protein intake of 1.2/g/kg/day and up to 1.5/kg/day if sarcopenia is present [90].

#### Supplementation of Protein and Bone Health

In a 2011 randomized study, which involved 219 postmenopausal women for two years, we wanted to verify whether supplementation with whey protein could have beneficial effects on bone structure; the sample was divided into two groups: the supplementation group took 30 g of whey protein, while the placebo group took a drink with the same energy content but with a protein intake of 2.1 g of protein. At the end of the two years of the study, the authors concluded by stating that protein supplementation brought neither benefits nor damage to the bone structure even if a significant increase in IGF-1 was highlighted in the group that received the supplementation [91].

A randomized double-blind study was conducted subsequently in 2015 for 18 months by Kerstetter and colleagues, in which 208 men and women with BMI between 19 and 32 kg/m^2^ and a self-reported protein intake of 0.6–1 g/kg/day were recruited; the control group took 45 g of whey protein as a supplement to the usual diet, while the placebo group took the caloric equivalent in maltodextrin. At the end of the 18-month study, there were no significant differences between groups for changes in BMD either in the spine or in other sites of skeletal interest [92].

A more recent work, but of very short duration, investigated the nocturnal administration of a supplement based on whey protein and fortified in calcium and vitamin D in order to evaluate the change in serum and urinary markers indicative of bone remodeling; the sample was represented by 16 postmenopausal women with an average age of 64.7 ± 3.3 years, an average weight of 67.2 ± 10.8 kg, and an average BMI of 26.0 ± 4.1 kg/m², divided between the control group and the group placebo. At bedtime, the control group took a supplement that included 0.3 g/kg/body weight of whey protein +0.3 g/kg/body weight of maltodextrin, while the placebo group took a drink that included 0.6 g/kg/body weight of maltodextrin. Bone-resorption markers were analyzed at time zero and at each hour in the following 4 h, and urine was collected for 24 h. The results showed a 30% reduction in CTX (C-terminal cross-linked telopeptide type I collagen) in the treated group compared to the placebo group, assuming that the intake of a serum protein product in the evening or before bed of milk can intervene on the homeostasis processes of bone tissue, helping to counteract the onset of degenerative bone diseases, such as osteopenia or osteoporosis [93]. Further studies are needed to evaluate the positive effects of a protein-based supplementation on bone metabolism.

In conclusion, adequate protein intake (at least 1 g/kg/body weight) is essential for maintaining adequate BMD and muscle mass and strength.

Given this background, the diet should contain foods with protein of animal origin (white meat: three portions/week, red meat: one portion/week, milk and yogurt: one portion/day, twice a week cheese, eggs: two portions/week, fish: four servings/week) and protein of vegetal origin (legumes: 2–3 portion/week) in order to prevent or treat BMD and muscle loss.

Finally, it is also interesting to consider foods that contain both animal proteins and calcium, so, in addition to milk and derivatives, fish should not be underestimated, where in some cases, as for smelts, a calcium content of 888 mg/100 g with a protein intake of 14.6 g/100 g is reached; just as good are the values of octopus, squid, and anchovies, which provide about 144–148 mg of calcium per 100 g of product with a protein quantity of 11 to 17 g/100 g of food. Almonds, among the group of nuts, also represent a good union between proteins and calcium, with contributions respectively equal to 6.6 g and 72 mg per 100 g of food.

### 4.6. Vitamin A

Vitamin A is a fat-soluble vitamin present both in foods of animal origin (for example, dairy products, liver, and eggs), as pre-formed retinol, and in foods of plant origin (fruit and vegetables) as provitamin A (especially α-carotene, β-carotene, β-cryptoxanthin) [94].

In the human body, 90% of vitamin A is stored by the liver in the form of retinyl esters, and it is transported in the serum as retinol linked to the retinol-binding protein (RBP); it is therefore probable that low levels of circulating retinol are indicative of vitamin A deficiency, while plasma retinyl esters would represent a more sensitive index of toxicity [95].

Carotenoids, otherwise called “provitamin A” (β-carotene in particular), are converted to retinol by the cells of the intestinal mucosa. Traditionally, 6 µg of β-carotene contained in foods are considered nutritionally equivalent to 1 µg of retinol, but the U.S. Institute of Medicine, based on studies on the efficiency of the conversion of provitamin A into vitamin A, has suggested a factor of 12 µg of β-carotene/retinol equivalents (RE) [95].

A recent review on current evidence by Yee et al., reported that studies on animal models have not shown any effect on bone of low doses of vitamin A, which, however, at high concentrations, seems to negatively affect bone formation, especially at the cortical level; β-carotene, on the other hand, has a potential protective effect against bone loss in mouse models [94].

The mechanism by which vitamin A could affect bone health has not yet been fully understood. Retinoic acid, an active metabolite of vitamin A, acts by binding to specific nuclear receptors and, in vitro, suppresses osteoblastic activity and stimulates the formation of osteoclasts [95]. It has also been hypothesized that this unfavorable effect of vitamin A may be linked to the alteration of the metabolism of vitamin D, as suggested by a 2001 study on nine healthy subjects, in which the administration of 15 mg of retinyl palmitate (equivalent to approximately one portion of liver) appeared to interfere with intestinal calcium absorption in response to physiological levels of vitamin D [96].

Already, a 2004 review conducted on 20 clinical studies highlighted how the intake of retinol through diet or supplements was negatively associated with lumbar and femoral BMD (neck and trochanter), with a progressive increase in relative risk of hip fractures with increasing retinol but not beta-carotene intake and with the occurrence of adverse events for retinol intake levels of approximately twice the recommended adult values (6.7 µg RE/kg body weight/day) [97,98]. The authors therefore concluded that, pending further research, the use of vitamin A supplements was not recommended for the purpose of improving bone health [97]. Similarly, an association between the risk of hip fractures and osteoporosis and a relatively low dietary intake of pre-formed vitamin A emerged from a 2006 review including four prospective observational population studies conducted in Scandinavia and the United States, equal to 1500 RE, considerably lower than the tolerable upper intake level (UL, 3000 RE), [99] which is why a quantity of 1500 µg/day has been suggested for the long-term intake of vitamin A (less than double the recommended levels for pregnant women, equal to 800 µg/day) [100]. Jackson and Sheehan, analyzing the results of six observational studies, also hypothesized a correlation between excessive intake of vitamin A (but not beta-carotene) and reduced BMD and increased risk of fractures [101] although a 2004 Danish case-control study on peri-menopausal women did not show any harmful effect of vitamin A on bone [102].

Finally, a recent review by Yee and colleagues analyzing the literature available to date on the effects of vitamin A on bone health found discordant results, probably due to differences in number, sex, and health status of the subjects investigated and duration and design of the published studies: of these, in fact, eight studies showed a protective effect of vitamin A against bone, 10 showed negative effects, while nine found no association between vitamin A intake and BMD or risk of fractures. However, some interesting considerations emerge from the analysis: the intake of vitamin A may not have harmful effects on the skeletal system in the presence of adequate serum levels of vitamin D; vitamin A could contribute to bone loss in overweight or obese subjects; finally, the potential negative effects of this vitamin could be counteracted by the concomitant intake/integration of other antioxidants or drugs with protective action (such as estrogens). Some of the studies included in the review, however, highlighted a positive correlation between beta-carotene and beta-cryptoxanthin (provitamin A) introduced through the diet and the BMD detected at various sites (including lumbar and femoral) and the risk of fracture, suggesting a protective role of provitamin A against bone [94].

In conclusion, therefore, based on the most recent literature, supplementation with vitamin A is not recommended in order to promote bone health, while the intake of provitamin A (especially β-carotene) through the diet seems potentially useful. Since in-vivo studies on provitamin A are few, optimal intake levels have not yet been established, while the recommended daily allowance (RDA) in adults for vitamin A is 700 µg for men and 600 µg for women not pregnant/breastfeeding [103]. Foods particularly rich in retinol include cod liver oil (18,000 μg/100 g), bovine liver (16,207 μg/100 g), river eel (1200 μg/100 g), butter (906 μg/100 g), pecorino (480 μg/100 g), tuna (450 μg/100 g), scamorza cheese (335 μg/100 g), buffalo mozzarella (240 μg/100 g), whole chicken egg (211 μg/100 g), and Greek yogurt (115 μg/100 g); those with a high content of β-carotene, on the other hand, include (quantities expressed in equivalent retinol) concentrated tomato (15,000 μg/100 g), carrots (6888 μg/100 g), valerian (4254 μg/100 g), sweet potatoes (3930 μg/100 g), yellow squash (3594 μg/100 g), mango (3198 μg/100 g), spinach (2910 μg/100 g), apricots (2160 μg/100 g), beets (1578 μg/100 g), and khaki (1422 μg/100 g) [104].

In the five portions of fruit and vegetables to be taken daily, orange-colored fruits and vegetables, due to their beta-carotene content, are therefore recommended foods in the diet of the subject with osteopenia/osteoporosis.

### 4.7. Vitamin D

For some time now, scientific literature has shown that vitamin D is an essential nutrient that plays a fundamental role in maintaining serum concentrations of calcium and phosphate and is essential for the development and maintenance of bone health throughout the life. The main effect of the active metabolite of vitamin D 1,25-(OH)_2_ D is to stimulate the absorption of calcium from the intestine. Vitamin D deficiency is very common in patients with osteoporosis. Vitamin D deficiency not only aggravates osteopenia and osteoporosis in both women and men but also causes a skeletal mineralization defect and muscle weakness, resulting in an increased risk of fracture. The main source of vitamin D for most humans is exposure to sunlight. Very few foods naturally contain vitamin D: oily fish, such as salmon and mackerel; cod liver oil; irradiated mushrooms; and egg yolk [3].

The RDA for vitamin D is 600 IU for all ages, with the exception of those over the age of 70, for whom 800 IU is indicated. According to the IOM committee, these quantities would guarantee a serum 25 (OH) D level of at least 50 nmol/L in 97.5% of the population, thus ensuring “good bone health” for most individuals [105].

Reasonable sun exposure (usually 5–10 min of arm and leg or hand, arm, and face exposure, two or three times per week) and an increase in dietary and supplemental vitamin D intakes are reasonable approaches to ensure sufficient vitamin D [3]. Factors affecting skin vitamin D production are season, latitude, time of day, skin pigmentation, aging, use of sunscreen, and glass [106,107]. The skin synthesis of vitamin D, in fact, decreases with increasing age; consequently, the percentage of people with vitamin D deficiency is higher in the elderly. Therefore, the dietary intake of vitamin D3 must increase in the elderly, but it is not easy to achieve this if the diet is not abundant in foods rich in vitamin D [108]. Finally, the statement of the American Academy of Dermatology must be highlighted, according to which UV radiation from the sun or artificial sources is a known carcinogen; therefore, it may not be safe or effective to obtain vitamin D through sun exposure.

A recent position statement investigating the current state of vitamin D in European and Middle Eastern countries highlighted a high prevalence of a low serum concentration of 25 (OH) D (<50 nmol/L), which reaches values over 50% during the winter season. Even more worrying is the presence of severe vitamin D deficiency (less than 25/30 nmol/L) in specific risk groups, such as children, pregnant women, and the elderly. The spectrum ranges from an adequate level of vitamin D in the Nordic countries to a severe deficiency in the Middle East. According to current evidence, the desirable serum concentration of 25 (OH) D is set at 50 nmol/L or higher. This can result in targeted approaches, such as prudent sun exposure, adequate nutrition, a food-enrichment policy, and vitamin D supplementation for high-risk groups [109]. It is important to ensure an adequate intake of vitamin D from a young age and throughout adolescence, especially through good sun exposure, given the low presence of vitamin D in the diet, thus facilitating the achievement of an adequate peak in bone mass [110]. A subclinical vitamin D deficiency is also linked to a greater risk of fracture and osteoporosis [111].

In support of the use of calcium and vitamin D supplements as an intervention for fracture-risk reduction in institutionalized middle-aged and elderly adults, the meta-analysis of randomized controlled trials of calcium supplementation (1000–1200 mg/day) and vitamin D (800 IU/day) by CM Weaver et al., highlighted a significantly reduced risk of 15% of total fractures and 30% of hip fractures [112].

It has therefore been shown that vitamin D supplementation, when administered with calcium, increases BMD and reduces the incidence of hip fracture in elderly subjects. Despite its widespread use, the benefits of vitamin D supplementation in younger women and as a single agent are less clear. A two-year randomized placebo-controlled double-blind twin study was carried out to measure the effect of vitamin D3 supplementation on bone density and bone metabolism in postmenopausal women. Seventy-nine pairs of women were recruited, and for each pair of twins, one was randomized to 800 IU cholecalciferol/day for two years, and the other was randomized to placebo. At the end of the study, no significant changes were observed in PTH concentration, serum calcium, or markers of bone resorption and formation, and no significant difference was observed for BMD at the spine or hip. Based on these results, supplementation of vitamin D alone cannot be routinely recommended as prevention of osteoporosis in postmenopausal women under 70 with normal vitamin D levels [113].

Additionally, some meta-analyses on vitamin D supplementation with or without calcium were carried out. A meta-analysis of Bishoff-Ferrari evaluated RCT’s with supplemental vitamin D in order to prevent falls among elderly individuals, for a total of 2426 subjects with a supplementation of ≤400 IU/day or 482 to 770 IU/day of vitamin D. It was possible to conclude that there was no fracture reduction for a received dose of 400 IU/d or less, whereas a higher received dose of 482 to 770 IU/d of supplemental vitamin D reduced nonvertebral fractures by 20% and hip fractures by 18% [114]. Reid’s 2014 meta-analysis then evaluated 23 studies (in eight of these, serological vitamin D levels were below the cutoff value of 50 nmol/L, and in 10 studies, participants were given vitamin D doses below 800 IU/day) with an average duration of 23.5 months; out of 4082 participants, 92% were female, with an average age of 59 years. In the studies examined, BMD was measured from one to five sites in the lumbar region, femoral neck, forearm, hip, or trochanter. Out of the total of the studies, only six reported a significant benefit and only two only a slight benefit, while the rest did not show significant changes. The meta-analysis therefore concludes that continued widespread use of vitamin D appears to be inappropriate for the prevention of osteoporosis in community-living adults without specific risk factors for deficiency [115]. According to Reid’s study, in 2017, Zhao’s meta-analysis analyzed 33 randomized trials in elderly adults, with a total of 51,145 participants, on the efficacy of calcium supplementation, vitamin D, or the use of both in reducing the risk of fracture, both total and on specific sites. In this meta-analysis of randomized clinical trials, the use of supplements that included calcium, vitamin D, or both versus placebo or no treatment was not associated with a lower risk of fractures among older adult community residents, thus not supporting the routine use of these supplements in this population sample [116].

A recent systematic review and meta-analysis evaluating fracture risks associated with differences in 25-hydroxyvitamin D (25 (OH) D) concentrations and fracture risks associated with supplementation with vitamin D alone or in combination with calcium found that neither daily nor intermittent dosing with standard doses of vitamin D alone was associated with a reduced risk of fracture, while daily supplementation with vitamin D and calcium was a more promising strategy. In fact, the meta-analysis of six randomized controlled trials on a total of 49,282 participants on the combination of both calcium and vitamin D supplementation showed a reduction in the risk of total fracture of 6% and a reduction in the risk of fracture of the hip by 16% [117].

Moreover, the assumption of vitamin D-fortified foods was investigated, as in the trial of Bonjour et al.: 37 institutionalized women, with a mean age of 84.8 years and with low serum 25-hydroxyvitamin D (mean: 5.5 ng/mL), were instructed to consume during one month two servings of soft, plain cheese made of semi-skimmed milk providing daily 2.5 microg vitamin D, 302 mg Ca, and 14.2 g proteins. At the end of the study, a reduction of bone-resorption markers was observed by positively influencing Ca and protein economy, as expressed by decreased PTH and increased IGF-I, respectively [118].

In conclusion, the current state of vitamin D in European countries shows a high prevalence of a low serum concentration of 25 (OH) D (<50 nmol/L), and several studies support the integration of vitamin D and calcium for the reduction in the risk of total fracture and a reduction in the risk of hip fracture. However, the dietary intake of vitamin D remains an important support for having healthy bone. The foods richest in vitamin D are listed as follows in descending order: cod liver oil (210 ug/100 g), herring (30 ug/100 g), salmon (17 ug/100 g), tuna (16.3 ug/100 g), catfish (12.5 ug/100 g), swordfish (11 ug/100 g), anchovies (11 ug/100 g), farmed sea bass (11 ug/100 g), grouper (11 ug/100 g), dogfish (9.10 ug/100 g), and chicken egg (4.94 ug/100 g) [104].

### 4.8. Vitamin E

Vitamin E is one of the most important antioxidants in the body, protecting against the effects of toxic radicals. Oxidative stress induced by vitamin E may interfere with the bone formation activity of osteoblasts, which in turn can lead to osteoporosis [119].

The two major members of the vitamin E family are tocotrienols and tocopherols. The most abundant form in human plasma is alpha-tocopherol. The current RDA for vitamin E is 15 mg/d, and the upper limit is considered to be around 800 to 1000 mg per day.

The review published in 2014 by Guralp collected studies on the effects of vitamin E intake on BMD, fracture risk, and bone-formation and -resorption markers in postmenopausal women. The author concluded by stating that although some benefits have been shown in observational studies, the current evidence does not prove a causal relationship between vitamin E and osteoporosis and the risk of hip fracture in perimenopausal women. Therefore, to better evaluate this association, intervention studies, both randomized and controlled (RCT), would be needed [119].

In the systematic review published by Shuid et al. in 2019, the studies in the literature concerning the therapeutic effect of vitamin E in preventing bone loss were analyzed. The studies published between 2005 and 2015 and examined were in total eight, of which five were on animals, one was in-vitro, and two were on humans. These studies showed that vitamin E, in particular tocotrienol, has an important anti-inflammatory and immunomodulatory role related to bone metabolism. It is through these mechanisms, the authors concluded, that vitamin E exerts its anti-osteoporotic action [120].

A meta-analysis published in 2020 showed that an increased intake of antioxidant vitamins, especially vitamin E, significantly reduces the risk of fractures at any site. In total, 13 prospective cohort studies were selected for a total of 384,464 individuals. The summary RR indicated that increased antioxidant vitamin intake was associated with a reduced fracture risk (RR: 0.92; 95% CI: 0.86–0.98; *p* = 0.015). When stratified by the vitamin types, increased vitamin E intake was found to be associated with a reduced fracture risk, whereas increased vitamin A and C intake did not affect this risk. Increased antioxidant vitamin intake was associated with a reduced fracture risk [121].

In order to investigate the relationship between the intake of various nutrients and BMD in middle-aged women, a cross-sectional study was conducted by Odai et al., published in 2019. Participants (157 women aged 38–76) were divided into premenopausal (n = 46) and postmenopausal (n = 111) groups. The correlation between the BMD Z-scores (measured with DXA) and the daily intake of 43 nutrients (assessed by food frequency questionnaires) was studied, and in the group of premenopausal women, a positive correlation was observed between alpha-tocopherol intake and BMD Z-score. Even correcting for age, BMI, smoking, alcohol, and exercise, daily vitamin E consumption remained significantly correlated with BMD. The authors concluded that an increase in dietary vitamin E intake could be an aid in maintaining bone mass in pre-menopausal women [122].

Finally, a recent Mendelian randomization study reconfirmed the existence of a causal association between increased circulating levels of alpha-tocopherol and improved BMD, measured with heel ultrasound and investigating the history of fractures. In inverse variance weighted analysis, a predicted one-standard-deviation increase of circulating alpha-tocopherol was associated with 0.07 g/cm^2^ in BMD, which corresponds to a >10% higher BMD [123].

In conclusion, although there are few studies in humans on the correlation between vitamin E and bone health, as this vitamin is a powerful antioxidant, it would exert a positive anti-inflammatory and immunomodulatory action on the bone, reducing the risk of fracture and development of osteoporosis.

Therefore, it is good to include foods containing this vitamin in the diet. As a fat-soluble vitamin, it is found mainly in vegetable oils, such as sunflower oil (49.20 mg per 100 g), sweet almond oil (45.80 mg), corn oil (34.50 mg), palm (33.10 mg), rice (32.30 mg), and extra virgin olive oil (21.42 mg); dried fruit, such as sweet almonds (26 mg per 100 g) or hazelnuts (24.98 mg); and all dried aromatic herbs (parsley, rosemary, sage, thyme, and marjoram all contain about 9.15 mg of vitamin E per 100 g), which are a great way to flavor dishes and limit the use of salt.

### 4.9. Vitamin K

Vitamin K is a fat-soluble molecule that can occur in two biologically active forms: vitamin K1 (phylloquinone) and vitamin K2 (menaquinone).

It is mainly known for its role in the blood coagulation mechanism, but it also has an important function in maintaining bone structure, acting as a cofactor of enzymes (vitamin K-dependent carboxylases) involved in the metabolism of this tissue [124] by increasing some markers of bone formation (such as alkaline phosphatase and IGF) through the differentiation of osteoblasts [125] and by regulating the mineralization of the extracellular matrix thanks to its osteocalcin carboxylation activity, which thus binds the hydroxyapatite crystals. These activities are mediated, in particular, by vitamin K2 [126].

Although vitamin K can be deposited in the extra-hepatic tissues of the liver, it cannot be stored in high quantities within the human body and therefore must be introduced in adequate quantities daily [127].

Regarding its role in bone metabolism, a 1999 study considered the relationship between dietary intake of vitamin K, assessed by food frequency questionnaire, and hip BMD (measured by DXA) in 72,327 women aged between 38 and 63 years. The results of the study showed that there was a positive correlation between vitamin K intake and BMD, as women who had reduced intake levels of this nutrient (<109 μg/d) also had a higher risk of fracture on the hip level (RR, with 95% CI, equal to 1.00) as compared to those in the highest quintiles (Q2 to Q5−intakes > 109 μg/d−a RR of about 0.7) [128].

A 2014 review analyzed four epidemiological studies to summarize the role of vitamin K on bone health in elderly subjects, discussing its clinical implications [129]. Among them, the Nurses’ Health Study, with 72,327 subjects, found an increased risk of hip fracture in nurses who had a lower-quintile vitamin K consumption (<109 μg/day) than those with consumption at the highest quintile. Similarly, the Framingham Heart Study showed that low intakes of vitamin K in 75-year-old men and women for more than seven years resulted in a significant association. The review, which includes a total of 77,410 adult patients (including 1853 men) with an age range between 32 and 86 years, seems to confirm that an adequate intake of vitamin K may be related to a reduction in the risk of fracture, but it is not associated with positive changes in BMD. Studies regarding vitamin K supplementation have evaluated the correlation with a possible improvement in bone metabolism. One of these, conducted by Inaba et al. in 2015, aimed to evaluate the minimum daily dose of menaquinone-7 (MK-7), which is biologically effective in improving osteocalcin carboxylation. Two studies were carried out: one with 60 postmenopausal women (aged between 50 and 59 years) allocated in four different supplementation groups (0, 50, 100 or 200 μg) for a duration of four weeks and the other study, on the other hand, with 120 subjects between the ages of 20 and 69, one-half of whom was assigned a placebo and the other a supplementation of 100 μg of MK-7. Both studies showed that MK-7 increased osteocalcin carboxylation in a dose-dependent manner, and significant effects could be observed both in the group that was given a supplementation of 100 μg and in that with 200 μg. This study shows how a supplementation of menaquinone-7 greater than 100 μg can improve the carboxylation of osteocalcin and, consequently, bone metabolism [130].

A similar study conducted in 2016 evaluated the intake of menaquinone-7 (MK-), in 375 μg tablets, on the bone metabolism of 148 postmenopausal women, also measuring BMD with the DXA method. The treated group compared to the placebo group had a reduction in non-carboxylated osteocalcin (ucCO), and it was also seen that the trabecular microarchitecture of the tibia, assessed with HRpQCT (High-Resolution peripheral Quantitative Computed Tomography), was maintained in treated subjects compared to the placebo group; on the other hand, no difference was found between the two groups in lumbar and femoral BMD, as assessed by the DXA method [131].

Another extremely recent study conducted by Morato-Martìnez aimed to evaluate the effect of 150 g of a preparation, similar to yogurt, enriched with calcium, vitamin D, vitamin K, vitamin C, zinc, magnesium, leucine, and probiotics administered daily for 24 weeks on bone metabolism markers in 65 postmenopausal patients between 50 and 60 years. Each participant was given guidelines for a healthy diet without, however, intervening on their usual consumption of dairy products and were subsequently randomly assigned to two groups: one group possessed the enriched preparation, while the other was assigned a similar preparation without the addition of other nutrients. The results of this study showed that the subjects who consumed the enriched product had higher levels of bone metabolism markers than the placebo group. This suggests that the consumption of a dairy product enriched in these nutrients (including vitamin K) may represent a primary prevention method against osteoporosis [132].

Finally, a 2019 meta-analysis by Mott carried out on 36 studies with a total of 11,112 participants who were followed for a follow-up ranging from 6 to 48 months stated that in osteoporotic or postmenopausal patients, fractures were lower in those who had vitamin K supplementation than in the control (2.24% and 3.06%, respectively, with OR of 0.72) but not at the level of vertebral fractures [133].

Vitamin K is an essential nutrient and must be constantly supplied to the body since inadequate intakes have shown a negative correlation at the bone level. The main food sources of vitamin K are green leafy vegetables, such as cabbage (440 μg/100 g) and spinach (380 μg/100 g), and vegetables belonging to the Brassicaceae family, such as broccoli (180 μg/100 g), as well as some fruits, such as kiwifruit (40.30 μg/100 g) and avocado (21 μg/100 g). If, therefore, a real efficacy of vitamin K from the point of view of BMD cannot be sustained, a diet rich in vegetables is highly recommended since it is beneficial from the point of view of fracture risk, as suggested by Shah’s review [129].

Therefore, a daily consumption of these foods sources of vitamin K is recommended in order to guarantee a constant supply of this nutrient for its involvement in the bone, particularly in age groups at risk of osteoporosis and in menopause women.

### 4.10. Vitamins B

From the in-vitro studies, it seems that an increase in homocysteine can promote osteoporosis through an increase in intracellular reactive oxygen species (ROS), which can increase the differentiation and the activity of osteoclasts. In this context, a deficiency of vitamin B12 could lead to a decrease in bone mass, [134] as suggested by an increase, respectively, of (3H)-thymidine uptake and of alkaline phosphatase activity [135].

A study conducted on Danish women found an association between femoral neck BMD and folate intake, but no correlations emerged between the intake of folate, vitamin B12, or vitamin B2 and the values of BMD at 10 years or in the risk of fractures [136]. Moreover, a 2013 Chinese study found that there was an inverse and dose-dependent correlation between vitamin B6 intake and hip fracture risk but only for female participants. In the same study, no significant association was found with the dietary intake of other B vitamins (thiamin, riboflavin, niacin, folate, cobalamin). The authors suggest that an adequate intake of vitamin B6 can prevent osteoporotic fractures in postmenopausal women and promote the health of bones, both directly on bone cell metabolism and indirectly through the modulation of steroid hormone receptors (including estrogen) [137].

Several intervention studies have been conducted on a possible relationship between supplementation with VB and bone health.

In a controlled double-blind trial, Hermann et al., showed that after one year of supplementation with a combination of folate, vitamin B12, and vitamin B6, there was no significant differences in femoral and lumbar BMD values in the experimental group, and only urinary DPD showed a highly significant reduction following vitamin supplementation. Only in those subjects with baseline homocysteine levels >15 μmol/L was the supplementation associated with a significant increase in lumbar BMD and a reduction in osteocalcin (OC) and N-terminal propeptide of type 1 procollagen (P1NP) [138].

A secondary study of the Women’s Antioxidant and Folic Acid Cardiovascular Study (WAFACS), conducted on women at high cardiovascular risk, found no difference in the risk of non-vertebral fractures and in serum markers of bone turnover (s-CTX and P1NP) after about 7.3 years of supplementation with folic acid, vitamin B6, and vitamin B12 [139].

A large, multicenter randomized controlled trial on subjects with moderately elevated homocysteine levels (B-PROOF study) reported that a supplementation of folic acid and vitamin B12 showed no significant effects on BMD or on QUS parameters. In this study there was a small but significant positive effect on a QUS parameter only in subjects aged >80 [140].

Green et al., (long-term randomized trial examining the effect of administering folate, vitamin B6, and B12 on bone biomarkers in elderly individuals with homocysteine levels >15 μmol/L) showed that after two years of treatment, there was no statistically significant difference in serum levels of bone-specific alkaline phosphatase (BSAP) and s-CTX [141]. Herrmann et al., (a small short-term study on folic acid supplementation) showed no effect of this supplementation on OC, P1NP, and s-CTX concentrations [142]. Similarly, Keser et al., and Shahab-Ferdows et al., found no significant differences in serum concentrations of bone turnover markers after a supplementation with folic acid and vitamin B12 and with vitamin B12 alone, respectively [143,144].

On the contrary, Carmel et al., observed an increase in alkaline phosphatase and osteocalcin levels after B12 supplementation but only in a small cohort of patients with vitamin B12 deficiency [145].

In a further randomized clinical study on a small number of postmenopausal osteoporotic women, a six-month supplementation with folic acid showed significantly lower serum osteocalcin levels, while there was a significantly greater increase in s-CTX and a lower concentrations of vitamin B12 in the placebo groups [146]. However, from a recent meta-analysis of seven randomized controlled trials, conducted with the aim of highlighting the possible existence of an association between treatment with vitamin B (folic acid and/or vitamin B12 and/or vitamin B6) and aimed at reducing the levels of homocysteinemia and fracture risk in adult subjects, found no statistically significant effect of the supplementation; this supplementation, therefore, is not recommended by the authors as a fracture prevention measure although the studies included in this meta-analysis have different dosages and combinations of vitamins. Furthermore, patients with cardiovascular diseases, colorectal adenomas, or hyperhomocysteinemia are included among the participants; therefore, it seems that there was many confounding factors that make it difficult to generalize these results [147]. In conclusion, these results suggest a possible impact of folic acid supplementation on bone metabolism rate, but the extent and the mechanisms involved remain unclear. and further research is needed to reveal any unknown aspects and confirm or refute this hypothesis [148,149]. In a study on dyslipidemic patients, after one year of treatment with 500 mL/day of skimmed milk fortified with EPA (eicosapentaenoic acid), DHA (docosahexaenoic acid), and oleic acid with or without vitamins A, D, E, B6, and folic acid, the group supplemented with vitamins showed a significant increase in osteocalcin, OPG, RANKL, and the OPG/RANKL ratio as well as an increase in serum folate concentrations, erythrocyte folate, and vitamin B6. These results suggest a positive effect of these supplementation on bone metabolism [150].

Similarly, Groenendijk et al., (randomized controlled trial on healthy subjects) showed an increase in bone formation after 24 weeks of supplementation with a fortified milk containing calcium, cholecalciferol, and vitamin B12 [151]. It is possible that this effect could also be due to IGF-1, which milk contains [152].

Moreover, Grieger et al., showed that six months of a daily multivitamin and mineral supplementation (including folic acid, vitamin B12, B6 and B2) is associated with a significant improvement in bone quality (greater increase for serum 25 (OH) D, folate, and vitamin B12; greater increase in QUS; trend towards a 63% lower mean number of falls) and with a reduction in the number of falls, which, however, did not reach statistical significance [153].

Another randomized clinical study investigating the effect of a supplementation with vitamin D and calcium with or without the addition of B vitamins (B12, B6, folic acid) on bone metabolism observed a significant and almost identical reduction in bone turnover in both groups, with a decrease in both formation markers and bone-resorption markers, suggesting that adding vitamin B to vitamin D and calcium supplementation does not lead to further improvements in bone turnover [154].

In conclusion, to date, the results emerging from literature about the potential role of vitamins B2, B6, and B12 in the promotion of bone health remain, while folic acid is recognized as having a pivotal role in maintaining bone health. Considering folic acid, all the studies that considered patients with hyperhomocysteinemia or with low blood levels of folic acid are in agreement, even if the number of patients is low, in demonstrating that folic acid supplementation is useful in improving BMD. Therefore, requesting folic acid and homocysteinemia dosage in elderly patients with osteopenia/osteoporosis is mandatory. For patients with hyperhomocysteinemia or with low blood levels of folic acid, all the studies are in agreement in demonstrating that folic acid supplementation (500 mcg–5 mg) is useful in improving BMD.

Particularly high concentrations of folate are found in brewer’s yeast (1230 μg/100 g) and in breakfast cereals (160–560 μg/100 g); among fresh vegetables and legumes, asparagus (211 μg/100 g), broccoli (125 μg/100 g), artichokes (120 μg/100 g), and spinach (110 μg/100 g) are particularly rich as well as broad beans (145 μg/100 g) and peas (65 μg/100 g); finally, various types of fruit also contain discrete levels of folate (oranges: 45 μg/100 g; kiwi: 28 μg/100 g; strawberries: 24 μg/100 g). As for food of animal origin, the highest contents are observed in liver and kidneys (chicken livers: 588 μg/100 g; bovine kidney: 98 μg/100 g) [103].

Vitamin B12 is naturally contained only in foods of animal origin and in some algae, while it is absent in all other foods of plant origin. The highest concentrations are found in offal (23 to 110 μg/100 g), fish (herring: 16 μg/100 g), shellfish (clams: 49 μg/100 g; average 19 μg/100 g), and crustaceans (crab: 9 μg/100 g; on average 3 μg/100 g) and in egg (yolk: 6.9 μg/100 g) and parmesan (4.2 μg/100 g) [103,155].

The highest concentrations of vitamin B6 in foods are found in beef (up to 0.66 mg/100 g), in cooked ham (0.61 mg/100 g), in salmon (0.75 mg/100 g), in tuna (0.49 mg/100 g), in chicken meat (0.23–0.51 mg/100 g), and in offal (0.1–0.8 mg/100 g) as regards foods of animal origin and in whole grain flours (content medium 0.4 mg/100 g), in nuts (pistachios: 1.70 mg/100 g; walnuts: 0.73 mg/100 g; hazelnuts: 0.59 mg/100 g), in legumes (dried lentils: 0.93 mg/100 g; dried chickpeas: 0.53 mg/100 g), and in vegetables (0.05–0.4 mg/100 g) as regards those of vegetable origin [103,155].

The highest concentrations of riboflavin in foods (present both in the plant and animal world) are found in offal (e.g., bovine liver: 3.3 mg/100 g); discrete quantities are then found in cheeses (camembert: 0.52 mg/100 g; pecorino: 0.47 mg/100 g; gruyere, fontina, cheddar: 0.45 mg/100 g), eggs (0.3–0.4 mg/100 g), milk (partially skimmed cow’s milk powder: 1.8 mg/100 g; partially skimmed pasteurized cow’s milk: 0.17 mg/100 g, variable according to the type of forage consumed by the cattle and seasonality), and green leafy vegetables (e.g., radicchio: 0.53 mg/100 g) [103,156].

### 4.11. Vitamin C

Many studies, both on animals and on cell cultures, showed that vitamin C may participate in osteoclastogenesis and osteoblastogenesis [157].

Moreover, studies on human subjects have shown that vitamin may affect bone mineral density and that a deficiency of ascorbic acid leads to the development of osteoporosis [158].

Recently, a 2020 meta-analysis [121] that assessed the relationship between fracture risk and intake of antioxidant molecules (including vitamin C) looked at 13 studies of prospective cohort, with a total number of subjects equal to 384,464. Of these studies, two meta-analyses [159,160] examined vitamin C alone, concluding that although there is no significant association between vitamin C intake and risk of fractures, a protective effect of ascorbic acid against hip fractures could be due to the stimulation of collagen synthesis type I and III, while its deficiency can stimulate osteoclastogenesis with consequent bone resorption.

Normally, the intake of vitamin C from the diet is sufficient to ensure optimal levels in the body; however, in the case of acute illnesses, the increase in demand can lead to a state of deficiency [161]. 

Taking into account elderly participants (334 men and 540 women, mean age 75 years, undergoing BMD evaluation by DXA and a 126-item FFQ), a negative association was described between the dietary intake of vitamin C, total and via supplements, and trochanteric BMD in smoking male subjects and a positive association between total vitamin C intake and femoral neck BMD in non-smoking males. A high overall intake of this vitamin was also correlated with a lower loss of femoral bone density during an observation period of four years in men with concomitant low intakes of calcium (<661 mg/day) and vitamin E (<7.7 mg/day); the attenuation of the significance of these associations following adjustment of the data for potassium intake (indicator of fruit and vegetable intake) suggested, however, that the effects of vitamin C may not be easily distinguished from those of other factors’ protective substances contained in foods of plant origin [162].

A recent narrative review [163] analyzed six studies (two randomized; one cohort; one observational; one randomized double-blind trial, and one randomized double-blind study) related to vitamin C supplementation and its correlation with bone metabolism, for a total of 2671 subjects examined. Most of the studies evaluated the supplementation of vitamin C in association with other nutrients (such as vitamin E), while only one study investigated the use of vitamin C alone as a supplement. In this study, which saw 994 postmenopausal women participate, only 277 regularly took the supplement that provided for the intake of vitamin C from 1000 to 5000 mg per day for a period of 12.4 years. The result was that BMD was 3% higher in the women who took the supplement. From the other studies examined, the vitamin C supplement was taken together with vitamin E, demonstrating a positive activity on bone metabolism, always evaluated by examining the BMD in some studies and, in others, the levels of vitamin D.

In conclusion, from the results of the various studies, it would seem that an adequate intake of vitamin C is not only favorable in improving the general health of the bones but can also act as a preventive tool against osteopenia and osteoporosis and fracture events.

Fruits and vegetables are the best sources of vitamin C. Citrus fruits, tomatoes (17.8 mg), and potatoes (18.2 mg) mainly contribute to the intake of vitamin C; other good sources are represented by red (128 mg) and green (80.4 mg) peppers, kiwi (74.7 mg), broccoli (89.2 mg), strawberries (56 mg), and Brussels sprouts (85 mg) [164].

### 4.12. Calcium

A systematic review published in 2017 investigated what the calcium intake was in various countries from a total of 78 studies covering 74 countries. Only some Northern European nations have shown that they have adequate calcium intake in their population, while the rest of the world is well below the recommended amount of 1000 mg/day [165].

The evaluation of the dietary calcium intake through a food frequency questionnaire of the femoral and lumbar BMD and of the fracture risk of an Italian outpatient population of 1000 subjects, of which the majority were women (838), showed that only 10.4% were taking adequate doses of calcium, i.e., higher than 1000 mg/day. The goal of this study was to evaluate the effects of calcium intake on BMD (evaluated both by DXA and by two ultrasound measurement techniques) and on the risk of fractures in the sample under analysis. Although no correlation was found between calcium intake and BMD, it emerged instead that changing from a calcium intake of <400 mg/day to an intake of >400 mg/day reduced the risk of fracture from 42 to 12%. Additionally, patients with one or more vertebral fractures had lower calcium levels (<400 mg/day) than those with no history of fractures. The authors therefore concluded by stating that, in order to counteract the risk of fragility fractures, it is good to recommend an adequate intake of dietary calcium [166].

A 2017 meta-analysis analyzing data from 17 studies for a total of 2537 individuals concluded that age and calcium intake are key factors affecting the efficiency and onset of BMD change. In particular, an intake for two years of 700, 1200, and 2000 mg/day, respectively, resulted in an increase in BMD of 68.0, 81.3, and 89.6%, confirming the fact that the dose of 1200 mg/day is already sufficient. The authors concluded by stating that a correct calcium intake can actually determine a reduction in the BMD loss curve in postmenopausal women, with a consequent reduction in the risk of fracture [167].

The 2018 systematic review by van den Heuvel analyzed the most recent articles with the highest reliability in the scientific literature, and the analysis of six meta-analyses out of a total of 33 randomized controlled trials and 25 prospective studies with a total of 426,595 subjects, concluded that the intake of 200–250 mL of milk per day (and therefore of a quantity of calcium between 240 and 300 mg) is associated with a reduction of 5% or more in the risk of fracture [168].

From this point of view, water is also an excellent source of calcium, useful not only for reaching the recommended daily quota but also able to contribute to the maintenance of BMD and therefore to the maintenance of bone health, as demonstrated in a review by Vannucci of 2018 [169]. The authors examined the studies in the literature on the role of mineral waters rich in calcium, as defined by 2009/54/EC directive (European parliament) and on their effect on bone metabolism.

Some of the studies examined agreed that the bioavailability of calcium from waters rich in this mineral is equivalent to or better than that of calcium contained in milk and derivatives. Most of the studies also demonstrate the positive effect of these waters on both bone biomarkers and densitometric parameters. In particular, clinical studies on postmenopausal women have shown that a regular intake of water rich in calcium significantly contributed to the maintenance of vertebral BMD in these women. The authors concluded by stating that water rich in calcium is an excellent, calorie-free source of highly bioavailable calcium and that it contributes to achieving the recommended daily intake of this mineral [169].

#### Supplementation of Calcium

In trying to define the role of calcium supplementation with or without vitamin D in the bone health in healthy male, in 2015, Silk and colleagues published a systematic metanalysis review. They analyzed six studies on 687 participants of ages between 16 and 84. They observed that the group with calcium supplementation with or without vitamin D presented better BMD values in young subjects and greater reduction of loss of BMD in older subjects than the control group [170].

A 2016 meta-analysis focused on the scientific literature to identify the benefits of calcium and vitamin D supplementation. The meta-analysis analyzed eight studies, for a total number of 30,907 adult subjects, and pinpointed the efficacy of calcium and vitamin D supplementation. Indeed, they showed that calcium + vitamin D therapy brings improvements in BMD and helps in reducing the risk of general fractures [57].

A further meta-analysis of six RCTs (49,282 participants, 5449 fractures, 730 hip fractures) of combined supplementation with vitamin D (daily doses of 400–800 IU, yielding a median difference in 25 (OH) D concentration of 9.2 ng/mL) and calcium (daily doses of 1000–1200 mg) found a 6% reduced risk of any fracture (RR, 0.94; 95% CI, 0.89–0.99) and a 16% reduced risk of hip fracture (RR, 0.84; 95% CI, 0.72–0.97) [117].

In conclusion, it can be inferred that the evaluation of calcium intake levels in subjects with BMD reduction via, for example, food frequency questionnaires, is of primary importance. Indeed, in those conditions of reduced intake, a diet that satisfies the daily calcium requirements needs to be followed.

Foods rich in calcium to be preferred and included in diet are cow’s milk (120 mg/100 g) and yogurt (125–150 mg), next, aged cheese (i.e., grana with 1162 mg /100 g or parmigiana with 1159 mg/100 g) and semi-mature cheese (i.e., scamorza with 512 mg /100 g or gorgonzola with 401 mg/100 g or pecorino with 470 mg/100 g); then, other calcium-rich foods include pilchards (613 mg), anchovies (542 mg), dried aromatic herbs (basil, 2110 mg; thyme, 1890 mg; sage, 1650 mg; origan, 1580 mg, etc.), tea in leaves (430 mg), dried figs (280 mg), and dried fruit (i.e., hazelnut with 150 mg/100 g and walnut with 131 mg/100 g). The consumption of 2 L of calcium-rich mineral water (when the calcium content is higher than 150 mg/L) is useful to reach the calcium daily requirements because it is valuable source of highly bioavailable calcium with beneficial effects on both bone biomarkers and bone densitometric parameters.

Thus, the diet should contain milk and yogurt: one portion/day, twice a week cheese, and 2 L/day of calcium-rich mineral water in order to prevent BMD loss.

Finally, if daily calcium requirements cannot be satisfied through diet, supplementation of calcium and vitamin D could be an effective strategy with a great benefit/cost ratio.

In the end, it is important to provide dietary guidelines to help prevent bone loss in stone-forming subjects. A diet based on an adequate intake of calcium (1000–1200 mg per day) and containment of salt can decrease significantly urinary supersaturation for calcium oxalate and reduce the relative risk of stone recurrence in hypercalciuric renal stone formers.

The restriction of dietary calcium can reduce the urinary excretion of calcium, but severe dietary restriction of calcium causes hyperoxaluria and a progressive loss of bone mineral component [171].

### 4.13. Phosphorus

Maintaining extracellular phosphorus homeostasis is essential for bone health as chronic phosphorus deficiency can cause bone demineralization and bone loss through resorption, according to a recent narrative review [172].

Prolonged dietary phosphorus deficiency can cause rickets and stunted growth in children and similarly osteomalacia in adults [172]; this is because phosphorus deprivation causes the skeleton to release calcium and hypercalciuria [173].

Phosphorus is a mineral that is particularly present in Western diets even in the form of a food additive, which is why cases of phosphorus deficiency are rare [174].

Several recent studies even show how excessive intake of this mineral as a mineral additive would seem to cause harmful effects to the bone structure [175] even in patients with physiological renal function [176].

A study by Gutièrrez and colleagues showed that the consumption of a diet rich in phosphorus, especially in the form of a food additive, increases the circulating concentrations of FGF23, osteopontin, and osteocalcin while statistically decreasing the concentrations of sclerostin in healthy individuals and in animals. Significantly, these values were also associated with substantial decreases in BMD in the mouse model. The study involved 12 subjects analyzed for two weeks and a sample of mice [176].

K.L. Tucker and colleagues, in a 2006 study, analyzed in a large cohort (1125 men, 1413 women) the usual dietary intake of foods and nutrients using a 126-item semi-quantitative food frequency questionnaire. Consistent and solid associations emerged between cola consumption and low BMD in women. The study results suggest that regular cola intake may help reduce BMD in women. Since BMD is strongly linked to fracture risk, and since cola is a popular drink, it is of considerable public health importance. However, these associations, which have previously been seen in teenage girls, remain controversial, and more research is needed. Although previous studies have implied that low BMD is due to dietary milk shift by carbonated beverages, we did not see any significant difference in milk intake by the cola-consuming group. Caffeine may help reduce BMD although low BMD remained after adjusting for caffeine intake, and some associations between low BMD and decaffeinated cola were also observed. The role of phosphoric acid on bone loss requires further investigation. Cola intake was associated with significantly lower BMD (*p* < 0.001–0.05) at each hip site but not in the spine and in women but not in men. The mean BMD of those on daily cola intake was 3.7% lower at the femoral neck [177].

The same authors in 2013 published a review in which it was considered how much the excess intake of phosphorus with the diet can contribute to the incidence of osteoporosis in the population. An example of food with added phosphorus is Coca-Cola, and 100 g of Coca-Cola contain 10 mg of phosphorus present as phosphoric acid. The habitual and chronic consumption of this drink is associated with osteoporosis and the risk of fractures [178]. According to authors Calvo and Tucker, future efforts by the scientific community should focus on a precise creation of databases of processed and phosphorus-added foods also showing the type of phosphorus source: natural or added or a combination of both. In fact, the absorption of phosphorus is greater for the added sources of phosphorus than the phosphorus naturally contained in food, and the phosphorus of animal proteins is more absorbed than that of vegetable proteins [178]. Furthermore, a better declaration and regulation on the use of phosphorus additives by the food industries would be essential to optimize the nutritional phosphorus avoiding toxic side effects, as also reported in a further recent review [172].

An important factor in assessing bone health would be the assessment of the ratio of dietary calcium to dietary phosphorus, as this ratio was found to be positively correlated with BMD in the femoral neck of men over the age of 50 and in premenopausal women [179].

Vorland, in a 2017 publication, also stated that the recommended calcium/phosphorus ratio is equal to 1: 1 and that a ratio shifted more towards phosphorus is therefore indicative of a low calcium intake, and an excessive intake of phosphorus may have effects negative on bone tissue [180].

Numerous studies in the literature therefore report how both excessively high and excessively low levels of dietary phosphorus can cause negative health effects and compromise the longevity of individuals. According to authors Juan Serna and Clemens Bergwitz, it could be important to consider the implementation of phosphorus as a routine measure in clinical practice [172].

Among the foods richest in phosphorus are worth mentioning as follows: wheat bran (1200 mg/100 g), sea bass (1150 mg/100 g), sea bream (1050 mg/100 g), cow’s milk (1030 mg/100 g), eggs (879 mg/100 g), seasoned cheeses (700 mg/100 g), flax seeds (660 mg/100 g), bitter cocoa (685 mg/100 g), black tea (630 mg/100 g), and mushrooms (612/100 g).

In conclusion, limiting consumption of inorganic phosphate additives is a strategic way to decrease phosphorus intake without affecting protein intake.

### 4.14. Magnesium

Magnesium is one of the minerals most represented in the human body after calcium, sodium, and potassium as well as the second intracellular cation after potassium [181]. Its blood concentration is finely regulated within a range from 75 to 95 mmol/L; despite this, several studies show that a concentration lower than 85 mmol/L is to be considered as a condition of deficiency. Within the body, magnesium reserves are represented by bone (53%), muscle (27%), and soft tissues (19%), while less than 1% is present in the serum.

It performs various functions in the form of a coenzyme, and among these, there is the conversion of vitamin D, a molecule responsible for the absorption and metabolism of calcium, in its active form (1,25-Dihydroxycholecalciferol), as well as a supporting role in normal functioning of the parathyroid glands, as illustrated by the systematic review by Schwalfendberg et al. in 2017 [182].

In particular, a narrative review [183] showed how reduced magnesium intake can negatively affect vitamin D levels in elderly subjects at risk of osteoporosis and in those at risk of deficiency, such as women, obese subjects, and/or with hyperparathyroidism [183].

Regarding the role of magnesium in bone metabolism, a 2016 in-vitro study showed how magnesium affects the activity of osteoblasts. A human osteoblast culture was treated with different magnesium concentrations (at concentrations of 1, 2, and 3 mM) for 24, 48, and 72 h, respectively. It was observed that the magnesium ions induced a significant increase in the viability of osteoblasts, the activity of alkaline phosphatase, and osteocalcin levels. Furthermore, osteoblast gap junctions were significantly promoted by magnesium. These findings may contribute to a better understanding of how magnesium affects bone remodeling [184].

Some studies have reported a positive correlation between magnesium intake and bone mineral mass, while others have found this association only for the femur and not for the lumbar vertebrae. Studies on a U.S. population show that 45% of Americans and 60% of adults do
not take adequate doses of this mineral [185,186]. This is certainly attributable to the eating habits of the population since although there are various causes behind the magnesium deficiency, a reduced dietary intake is its main cause. The processing processes that foods undergo negatively impact the amount of magnesium present in them [187]. Furthermore, according to various studies carried out, fruit and vegetables in the U.S. and the United Kingdom have lost 80% to 90% of their magnesium content in the last century [188]. Added to this is the difficulty of reaching a precise diagnosis of magnesium deficiency due to the multitude of factors that influence its status [187]. According to Razzaque’s narrative review carried out on two meta-analyses with a total of 553,098 subjects, a high number of cardiovascular, metabolic, neurological, respiratory, and bone pathologies are associated with a low intake of magnesium. In fact, among the diseases most often associated with an inadequate intake of magnesium are osteopenia and osteoporosis [189]. A 2015 observational study evaluated the intake of various minerals, including magnesium, in 51 postmenopausal women with an average age of 57.97 ± 1.2 years, average height of 156.8 ± 0.79 cm, and an average BMI of 27.62 ± 0.39 kg/m^2^, of which 23 had osteoporosis, and 28 had osteopenia. The results showed that the intake of various minerals, including magnesium, was lower than the recommended doses for age and physiological state [190].

A 2014 prospective cohort study evaluated magnesium intake as a risk factor for osteoporotic fractures and altered BMD. A total of 73,684 women enrolled in the “Women’s Health Initiative Observational Study” and aged between 50 and 79 years were evaluated. It was observed that women who consumed more than 442.5 mg of Mg per day showed a 3% higher BMD in the hip, while that in the whole body was 2% higher compared to those who took less than 206.5 mg per day of magnesium.

Conversely, a 2014 study instead showed a correlation between magnesium intake greater than 442.5 mg per day and the increased risk of forearm or wrist fractures [191].

Another cohort study from 2017 wanted to evaluate the protective effect of a correct intake of magnesium on the risk of fracture. Of the 3765 participants, 2071 women and 1577 men, with an average age of 60.8 years and an average intake of magnesium of 295 ± 116 mg/dL/day, only 27% of them had an adequate intake of the element. The group was later divided into quintiles based on the intake of magnesium. After an average period of 6.2 years, 560 individuals—362 women and 198 men—suffered a fracture. The quintile with the lowest intake of the mineral showed a higher fracture incidence as opposed to the quintile with the highest intake, which showed a lower fracture incidence. This underlines how a greater intake of magnesium would seem to have a protective role with respect to the risk of fracture [192].

It was demonstrated that ethnic differences can also influence magnesium intake, as in the study of Jackson et al., in which the association between ethnicity (Caucasian/African American/Hispanic/other) and magnesium intake was observed in a large representative sample of U.S. older adults (*n* = 5682). Specifically, magnesium intake remained lower among African American older adults (13.0 mg/day) and higher among those from other ethnic groups (17.2 mg/day) compared with Caucasian older adults. In addition, a higher intake of magnesium was observed among Hispanic older adults (14.0 mg/day) relative to Caucasians [193].

A 2006 study evaluated patients with osteoporosis, low blood levels of vitamin D, and high levels of PTH to verify the correlation with magnesium deficiency: 30 women over 60 years of age with osteoporosis and vertebral fractures, following physical and biochemical assessments, were recruited into the study and monitored for a period of 12 months, of which 29 completed the study. Following magnesium supplementation with a concentration of 2.4 mg/kg (0.1 mmol/kg) of magnesium sulphate in 50 mL of 5% dextrose infused over 4 h, it was possible to correct the deficiencies related to the levels of vitamin D and PTH, demonstrating as well as the lack of magnesium as a determining factor [194].

Finally, a 2009 study showed how oral magnesium supplementation in osteoporotic and postmenopausal women reduces bone turnover. A group of postmenopausal women with a T-score equal to or less than −2.5 was recruited to evaluate the effect of oral supplementation of magnesium (equal to 1.83 mg per day) for over a month on bone-health-related markers. The study, randomized and controlled, was carried out on 20 subjects, with a group of 10 women who were given the supplementation and a control group of the same number with similar characteristics (age, BMI, and duration of menopause) of the group that was taking magnesium. Within 30 days of supplementation, a significant reduction in PTH (from 6.17 ± 1.99 to 4.18 ± 1.36, *p* < 0.05) and an increase in serum osteocalcin levels (*p* < 0.001) was observed [195].

In conclusion, magnesium carries out various activities within the body; in particular, as regards bone metabolism, it represents a nutrient of primary importance for the normal development of the reactions involved in bone and in the hormonal control of its reabsorption, acting on vitamin D and PTH.

Among the dietary sources of magnesium there is primarily water, which contributes 10% of the daily magnesium requirement [181]; in addition to this, sources of magnesium are represented by green leafy vegetables, such as spinach (a portion of 200 g contains 185 mg), as well as almonds and cashews (270 mg per 100 g) and pumpkin seeds (550 mg per 100 g) [196].

### 4.15. Iron

Iron is an essential element involved in oxygen metabolism, and it is incorporated into proteins, such as hemoglobin present in the blood and myoglobin in muscle tissue, and acts as a cofactor for numerous enzymes involved in collagen synthesis and genic expression control [197].

As a component of all living cells, it is considered an essential micronutrient for human health: its deficiency can cause anemia and, consequently, fatigue, weakness, and dysfunction of the immune system, while its accumulation is related to an increase in risk of neoplastic diseases, heart failure, and diabetes mellitus [198].

To date, the relationship between iron and BMD is under study. In mouse animal models, a strong restriction in the iron administered with food was associated with a decrease in the number and thickness of bone trabeculae [197].

As for humans, a longitudinal study on a large population sample in Taiwan in 2017 identified iron deficiency anemia (IDA) as an independent risk factor for the development of osteoporosis [199]. However, since hypoxia and iron metabolism are closely linked, it is difficult to clarify the mechanism by which IDA negatively affects bone health [197]. In a study of otherwise healthy women with iron deficiency (serum ferritin ≤ 30 ng/mL), iron deficiency was however related to a greater degree of bone resorption, assessed by assaying the aminoterminal collagen I telopeptide, NTx [200].

On the other hand, the severity of iron overload, present in a sample of patients with hereditary hemochromatosis, was also independently associated with BMD measured at the femoral neck and lumbar spine, with the finding of a condition of osteoporosis in 25% of patients and about 41% of osteopenia [201].

A recent study in 2020 conducted on 4000 women aged between 12 and 49 years instead highlighted the existence of a negative linear association between serum levels of ferritin and BMD, while no correlation of dietary intake emerged for iron (assessed by two 24-hour recall) and sideremia with femoral and lumbar BMD (assessed by DXA) [198]. The association between serum ferritin and BMD had already been highlighted in other studies, such as the Korean National Health and Nutrition Examination Survey on a sample of 7300 women but only for premenopausal subjects (and not for postmenopausal ones) and limited to lumbar BMD [202].

It can therefore be concluded that an alteration both in defect and in excess of the state of iron can be unfavorable for the purpose of maintaining bone health, thus reiterating the importance of taking the recommended quantities for this mineral through the diet, equal to 10 mg/day for adult men and postmenopausal women and 18 mg/day for women of childbearing age [103].

The main dietary sources of iron in the form of heme iron (more bioavailable, of animal origin) include bovine spleen (42 mg/100 g), pork liver (18 mg/100 g), clams (14 mg/100 g), pheasant (8 mg/100 g), hare (6.2 mg/100 g), horse (3.9 mg/100 g), anchovies (2.8 mg/100 g), bresaola (2.4 mg/100 g), veal (2.3 mg/100 g), and lamb (1.7 mg/100 g) [104].

The sources of non-heme iron (of vegetable origin, characterized by lower bioavailability) include cumin seeds (66.4 mg/100 g), cinnamon (38.1 mg/100 g), dried mushrooms (28.9 mg/100 g), black pepper (28.9 mg/100 g), wheat bran (12.9 mg/100 g), herbs (leaves) (12.5 mg/100 g), wheat germ (10 mg/100 mg), dried chickpeas (6.1 mg/100 g), dried peas (4.5 mg/100 g), and roasted peanuts (3.5 mg/100 g) [104].

### 4.16. Copper

In-vitro and animal model studies have demonstrated the role of copper as a cofactor in bone metabolism, and therefore, an inadequate intake of this mineral can consequently be associated with the development of osteopenia or osteoporosis, as reported in the 2013 Zofková review [203].

In men, an observational study published in 2009 by Chaudhri analyzed the relationship between the concentration of copper in the blood and BMD (assessed with DXA) at the lumbar level in 25 osteopenic women in menopause. The results showed a statistical correlation between low plasma levels of copper and a reduced BMD, demonstrating the important role that copper plays in maintaining correct bone density [204]. A 2014 meta-analysis conducted by Zheng wanted to better understand, through the review of the current literature, the relationship between copper and other minerals with BMD. With regard to copper, in particular, five articles were analyzed, including a review, which included a total of 10 case-control studies. The meta-analysis of these 10 studies, involving a total of 830 adult subjects, led to results similar to those of the previously cited studies: patients with osteoporosis had lower serum copper levels than patients in the control groups. The meta-analysis ended by stating that low blood copper levels could be defined as risk factors for developing osteopenia or osteoporosis [205].

Contrasting results were obtained instead in a study conducted in 2014 by Sadeghi et al., out of a sample of 135 Iranian women. The results showed that the plasma level of some minerals (including copper) did not differ significantly between patients with osteopenia/osteoporosis (T-score < −1) compared to controls (T-score > −1) [206]. Finally, in 2018, a cross-sectional study was published that evaluated the association between plasma levels of copper and BMD at various sites and fractures (investigated through questionnaires) in a sample of 722 subjects, both men and women, enrolled starting from data from the National Health and Nutrition Examination Survey (NHANES) 2011–2014. The results of this study showed that the lowest BMD values were those of patients with the lowest plasma copper values, but at the same time, those with the highest copper values had a higher risk of fractures, especially in male patients [207]. The authors justified this latest evidence by stating that an excess of copper can produce large amounts of free radicals, which, by interfering with bone metabolism, can lead to a loss of cortex and strength. This would therefore explain the increased risk of fractures in subjects with high plasma levels of copper [207].

Regarding the studies that evaluate copper supplementation (dietary supplement or high intake) in relation to bone metabolism, in two studies, the individual effects of copper were assessed, [208,209] while two other papers evaluated the effects of a micronutrient association [210,211].

The two studies that analyzed the integration of copper without other micronutrients (2.5–3 mg/day) showed good results in terms of slowing down bone mineral loss and reducing reabsorption markers. The study published by Nielsen in 2011 (a double-blind placebo-controlled study) instead produced results that are not completely clear. Subjects (224 postmenopausal women with similar femoral neck T-score and BMI) were divided into two groups: one group (*n* = 112) was given a daily supplementation with 600 mg Ca plus placebo and the other given 600 mg Ca plus 2 mg Cu and 12 mg Zn for a period of two years, evaluating the BMD with DXA. The results showed that after two years, women with Ca plus Cu and Zn supplementation showed a significant decrease in BMD and T-score, while calcium supplementation alone did not show this decrease. The authors concluded that the negative effect is attributable to Zn, but that the reason has yet to be clarified. Food diaries also indicated that Cu intakes <0.9 mg/d were associated with decreased DXA bone status measurements, which suggests that long-term low intakes of these elements may increase the risk of osteoporosis [210].

In conclusion, it would seem that the status of copper in the blood is related to BMD. In particular, from most of the studies conducted, it emerged that low plasma levels of copper correspond to low levels of BMD and therefore a greater risk of experiencing loss of BMD. To prevent the loss of BMD, it is therefore useful to take the amount of copper equal to the RDA, which for adult men and women is 900 μg/day (Food and Nutrition Board, 2001); thus, foods with a higher content of copper are sheep liver (8.70 mg); bovine and horse liver (3.70 mg); oysters (7.50 mg); lamb (4.61 mg); herring (4.10 mg); unsweetened cocoa powder (3.90 mg); all dried fruit, especially cashews (2.00 mg); chestnuts (1.88 mg); tea leaves (1.60 mg); and squid, cuttlefish, and mussels (1.20 mg) [104].

In the event that food intake is not adequate, dietary supplementation with 2.5–3 mg/day is useful.

### 4.17. Zinc

Zinc is not only a component of bone (most of the zinc in the human body is stored in the bones) but also an essential cofactor of many proteins involved in microstructural stability and bone remodeling [212]. On the one hand, zinc stimulates osteoblasts, activating the processes that lead to the formation of bone tissue and promoting the production of collagen; on the other, it inhibits the bone-resorption processes carried out by osteoclasts [203].

Regarding zinc intake or serum levels, a position paper of Lowe et al., observed that some elderly groups may have reduced dietary intakes and/or absorption and that dietary supplements of zinc and other minerals can effectively reduce the rate of bone loss [213].

A recent meta-analysis showed that dietary zinc intake and serum zinc levels could play an essential role in osteoporosis prevention; in the overall analysis performed, the serum zinc level was significantly lower in patients with osteoporosis than in controls [214].

Mutlu’s study compared the blood levels of zinc of 120 postmenopausal women, subsequently divided into three groups of equal number (healthy, osteopenic, and osteoporotic). The results showed that mean zinc concentrations were significantly lower in osteoporotic women than in osteopenic or normal women. Furthermore, zinc concentrations in osteopenic women were significantly lower than in normal women [215].

The status of zinc within the human body could therefore be associated with changes in the bone tissue, just as occurs in states of osteopenia or osteoporosis. In the 2015 Mahdavi-Roshan review, it was highlighted that the levels of zinc in the blood, on average equal to 67.4 ± 3.32 μg/dl in patients including a total 51 women with an average age of 59 years and with osteopenia and osteoporosis, are significantly lower than the normal range [190].

Evaluating the effectiveness of zinc supplementation, a double-blind randomized study from 2013 performed by Mahdavi-Roshan aimed to examine the effect of zinc supplementation on serum zinc and calcium concentrations in postmenopausal osteoporotic women. The participants, in total 60 with an average age of 58.2 ± 1.2 years, an average weight of 68.0 ± 1.0 kg, and an average height of 156.8 ± 0.8 cm, were divided into two groups: the intervention group to which 220 mg/day of zinc sulphate were administered and the placebo group. Food intakes and blood levels of zinc and calcium were evaluated at the beginning and at the end of the study. At the beginning, mean serological levels of zinc were significantly lower than the normal range, while no significant difference in serum calcium concentration was found. After 60 days, the group taking zinc sulfate supplements showed a significantly higher serum zinc concentration than the initial zinc concentration, while there was no significant difference in serum calcium concentration [216].

An interesting implication inherent in zinc intake and its impact on osteoporotic condition was reported in the Rancho Bernardo Study (RBS), a population-based prospective study of older residents of a southern California suburban community. The aim was to examine the independent association between dietary zinc and plasma zinc and the association of each with BMD and four-year bone loss in community-resident elderly men. At the beginning of the study, data on food intake were collected using a standard questionnaire on the frequency of foods, and plasma zinc concentrations were measured using inductively coupled plasma spectroscopy. Of the original study subjects, 396 men (aged 45–92 years) completed BMD measurements in 1988–1992 and again four years later. Over the four-year interval, eighty-one men reported taking vitamin and mineral supplements that included zinc; in this subgroup, zinc intake supplementation averaged 27.5 mg/day. The total daily intake of zinc from the diet and supplements reached an average of 17.1 mg, and the overall average plasma concentration of zinc was 12.7 mol/L. The results showed a correlation between plasma zinc and total zinc intake (diet plus supplements); dietary zinc intake and plasma zinc concentrations were lower in men with osteoporosis of the hip and spine than in men without osteoporosis at those sites. The BMD for the hip, spine, and distal pulse were significantly lower in men in the lowest plasma zinc quartile (11.3 mol/L) than in men with higher plasma zinc concentrations. The association between plasma zinc and BMD was transverse, longitudinal, and independent of age or body mass index. However, plasma zinc did not predict bone loss over the four-year interval. The study concluded that dietary zinc intake and plasma zinc each have a positive association with BMD in men [217].

In conclusion, bearing in mind the role of zinc in bone metabolism and the reported studies, we can state that the serum level of zinc is significantly lower in patients with osteoporosis, and a supplementation of the same can be positively correlated with BMD. From a nutritional point of view, it is advisable to vary the consumption of foods that contain this mineral; in descending order from 100 g of product, the most zinc-rich foods are as follows: oysters (45 mg), wheat germ (17 mg), wheat bran (16.20 mg), trout (14 mg), mushrooms (14 mg), chicken egg (5 mg), turkey (4.9 mg), and nuts (4.53 mg) [104]. It would be useful in osteopenic and osteoporotic patients to measure the blood zinc in order to consider a supplementation that the literature recommends equal to 27.5 mg/day.

### 4.18. Silicon

Silicon is an element linked to glycosaminoglycans that is important for the formation of cross-links between collagen and proteoglycans [218]. In animal models, silicon showed the major concentrations in bone and other connective tissues [219], in particular in the immature osteoid tissue, suggesting a potential role in its mineralization and calcification [220].

In humans, silicon is absorbed mainly through the diet as orthosilicic acid (its most bioavailable form, present in liquids, such as drinking water and beer) or as polymeric and phytolytic silica (in foods); in its circulating form (silicic acid), silicon is not bound to proteins, and it is excreted by the renal tubules [221,222,223].

A recent narrative review by Rondanelli and colleagues including eight clinical trials (10,012 subjects) investigated the relationship between dietary intake and supplementation of silicon and BMD [224]. In men and in pre-menopausal women, silicon intake with the diet seemed to be positively related to femoral BMD (but not with lumbar BMD), with the major difference between the lower (<14 mg/d) and the upper (>40 mg/d) quintile of dietary intake, and the Si dietary intake was responsible for 0.1% of changes in BMD in peri-menopausal women. Moreover, there was a negative association between the intake of silicon and some urinary bone-resorption markers and a positive association with PINP, a serum bone formation marker; in a sample of men, in particular, the intake of Si through vegetables showed a positive correlation with total serum alkaline phosphatase, another marker of bone formation [224]. Moreover, the supplementation of 6 mg of silicon (in the form of orthosilicic acid stabilized with choline) in combination with calcium and vitamin D seemed to have a beneficial effect on bone turnover and bone collagen compared to the use of calcium and vitamin D alone [224].

In conclusion, even if at present an adequate level of intake (AI) and a tolerable upper intake level (UL) for silicon have not yet been established [225,226], considering the data from studies on animal models and humans, it has been suggested a dietary Si intake of about 25 mg/day in order to promote beneficial effects on bone health [224]. This intake can be reached through the consumption of those foods and drinks rich in silicon, in particular products of vegetables origin, including rice flour (565.40 mg/100 g), oats (425.00 mg/100 g), green beans (43.90 mg/100 g), crackers (13.00 mg/100 g), hazelnuts (10.00 mg/100 g), cauliflower (8.40 mg/100 g), tomato (6.10 mg/100 g), beer (6.00 mg/100 g), peanuts (5.80 mg/100 g), bread (4.78 mg/100 g), banana (4.77 mg/100 g), and lentils (4.42 mg/100 g) [227,228].

### 4.19. Manganese

To date, the available data are insufficient to be able to establish an average requirement (AR) and a reference intake of manganese for the population (Population Reference Intake, PRI). For European countries, however, the EFSA has proposed an adequate AI for adults equal to 3 mg/day [229]. Furthermore, currently, again due to the scarcity and inadequacy of available data, a tolerable UL for manganese has not yet been established [103,226].

Manganese plays the role of coenzyme in numerous biological processes, including the process of bone formation, as reported in the review by Erikson and Aschner [230]. Two previous reviews reported that in-vitro and animal model studies have shown that at the bone tissue level, manganese has the role of stimulating the synthesis of the bone matrix [203], as it is, in fact, a fundamental mineral for the incorporation of calcium and phosphorus into the bone tissue. In the absence of manganese, the bone structures would be more porous, more fragile, and with a lower bone density [231].

In 1987, in Friedman’s study, seven subjects underwent an initial 21-day period characterized by a baseline diet with a daily intake of Mn of 2.59 mg, then underwent 39 days of a “depletion period” with 0.11 mg Mn/day, and after 10 days undertook a progressively higher daily intake of Mn (1.53 mg the first five days and 2.55 mg/day for the last five days). A significant increase in serum concentrations of calcium, phosphorus, and alkaline phosphatase was found in the 39 days of a low-Mn diet, suggesting that Mn deficiency could affect bone remodeling [232].

In 1996, Itokawa demonstrated that bone alterations and bone loss were plausibly related to a lack of manganese intake in 15 patients undergoing total injecting nutrition [233].

In 1993, Saltman P.D. and Strause L.G. analyzed 59 osteoporotic women who showed significantly lower serum manganese levels than sample healthy subjects (0.02 vs 0.04 mg/L, respectively) [234].

A cross-sectional study from 2009 conducted on a group of 41 postmenopausal women revealed a positive correlation between dietary manganese intake and BMD and a negative correlation between manganese intake and number of fractures, thus meaning that this element can contribute to overall bone health and can be considered a predictor of bone status [235]. Subsequently, in a study conducted in 2016, significantly lower serum levels of calcium, manganese, copper, and zinc were found among osteopenic and osteoporotic patients compared to healthy controls. In addition, significant increases in serum calcium and manganese concentrations and decreases in copper and zinc were observed after a 12-week period of moderate aerobic physical activity. These changes were related to the increase in serum bone alkaline phosphatase levels and with the improvement of both femoral and vertebral BMD values. These observations seem to confirm the essential role of some trace elements, including manganese, in the synthesis processes of cartilage and collagen and in bone mineralization [236].

On the contrary, a 2020 cross-sectional study by Li Defu highlighted how retired workers who were exposed to high doses of manganese during work may have a higher risk of developing osteoporosis than the female control sample population and compared to male workers exposed to the same doses of manganese [237]. This study involved 304 subjects exposed to high doses of manganese for more than 10 years (161 men and 143 women) and 277 control subjects who were retired workers, including 65 men and 212 women. This was the first study to investigate the relationship between manganese exposure and the quality of bone density in retirees who were exposed to high doses of manganese during their working life.

Regarding manganese supplementation, a clinical trial conducted in Mexico in 2001 demonstrated how a multiple supplement with vitamins and microelements, including manganese (administration of a drink containing the recommended daily dose, RDA, of vitamins D3, E, K1, B6, niacin, thiamine, biotin, folic acid, and pantothenic acid and minerals iodine, copper, manganese, and selenium; 1.2 times the RDA of vitamin A and 1.5 times the RDA of ascorbic acid, riboflavin, vitamin B12, iron, and zinc), given six days/week for an average period of 12.2 months, was able to improve growth in length (with a gain of almost 5 mm) in children aged between 8 and 14 months compared to the placebo group. In particular, the effect of supplementation was greater in children less than 12 months of age at the start of the study (average increase in length of 8.3 mm) compared to children ≥ 12 months (average increase of 2.0 mm) [238].

In a prospective placebo-controlled double-blind study, Saltman and Strause evaluated the importance of calcium supplementation (1000 mg) with and without the addition of copper (2.5 mg), manganese (5 mg), and zinc (15 mg) (referred to as TMIN). The subjects were divided into four groups: placebo, supplementation with Ca only, supplementation with TMIN alone, and supplementation with Ca+ TMIN. Bone loss was significantly greater in the placebo group and in the TMIN-only supplementation group compared to the Ca+ TMIN group [234].

A very similar study was subsequently conducted by the same authors on a smaller group of postmenopausal women, always divided into four groups. After a period of two years, the group that received the Ca+ TMIN supplementation proved to be the only one to differ significantly from placebo with regard to the change in lumbar bone density. These results highlighted how the lumbar bone loss observed in postmenopausal women receiving calcium supplementation can be further slowed through the simultaneous increase in the intake of trace minerals [211].

Given the evidence, manganese would appear to be an important element which, if not taken in adequate doses, would play a role in the complex pathogenesis process of osteoporosis and which is very often not evaluated or considered.

Among the foods rich in manganese, we must mention black tea leaves (73 mg), nuts (especially pine nuts, 7.90 mg; hazelnuts, 4.40 mg; walnuts, 3.40 mg; peanuts, 2.10 mg; pistachios, 1.20 mg), cocoa (2.50 mg), coffee (2.10 mg), coconut flour (1.80 mg), peanut butter(1.70 mg), trout (1.09 mg), perch (0.70 mg), shellfish (clams, 1.0 mg), and some fresh fruit (peaches, 0.80 mg; figs, 0.50; chestnuts, 0.50 mg; pineapple, 0.50 mg; banana, 0.40 mg; apricot, 0.40 mg; raspberries, 0.40 mg).

### 4.20. Boron

Humans absorb boron primarily through inhaled air, food, and water and minimally through intact skin [239,240]. Physiology studies demonstrated that boron tends to reach higher levels in bone tissue (mineral portion), liver, lymph nodes, adrenals, and kidneys [241,242], and it is excreted mainly through urine [243].

Because of the lack of data in literature, there is no recommended intake levels (RDA) for this mineral, but the WHO has set an acceptable safety level for boron intake of 1–13 mg/day [244].

Studies in animal model and in vitro demonstrated that boron plays an important role in metabolism of calcium, 1,25-(OH)_2_ vitamin D, testosterone, and 17β-estradiol, being necessary for the hydroxylation of the steroid ring, as reported in a critical review by Deviran [245], and it seems to play a role in improving the absorption and bone deposition of magnesium, which contributes in the regulation of calcium metabolism, as reported in another review [246]. In humans, a study by Boyacioglu et al., demonstrated that a natural exposure to higher levels of boron (drinking water “rich in B”: 1.59 ± 0.04 mg/L; average daily intake: 6.99 ± 2.90 mg) appeared to be correlated with higher serum levels of osteocalcin (an indicator of osteoblastic function) in 25 post-menopausal women (50–60 years) compared to those (*n* = 28) coming from regions “poor in B” (drinking water with 0.012 ± 0.05 mg/L of boron; average daily intake: 1.20 ± 0.12 mg) [247].

A recent narrative review including seven clinical trials (594 subjects) showed the effects of boron supplementation on bone health: a boron supplementation of 3 mg/day in post-menopausal women was associated to a reduction of the urinary excretion of calcium and magnesium and to an increase of the serum levels of 17-β estradiol and testosterone as well as a daily supplementation of 10 g of sodium tetraborate in a sample of healthy men, while a diet low in boron (0.33 mg/day) was associated to an increase of urinary calcium excretion. Moreover, the effect of boron on the excretion of calcium seemed to be influenced by the eventual intake of magnesium [248]. Moreover, also the supplementation of boron in addition to other nutrients (calcium, strontium, magnesium, vitamin D3, vitamin K, vitamin C) showed an association with the improvement of bone health, with a positive increase in BMD measured by DXA [248].

In conclusion, even if at present an RDA for boron has not yet been set, achieving an intake that is within the safe range, as suggested by WHO (1–13 mg/day), is essential for bone health.

Reaching this intake of boron is not always easy with the diet, as boron can be found mainly in drinking water and some foods, such as sultanas (2200 µg/100 g), peanuts (1700 µg/100 g) and peanut butter (1450 µg/100 g), wine (610 µg/100 g), peaches (530 µg/100 g), grapes (490 µg/100 g), peas (460 µg/100 g), apples (360 µg/100 g), pears (280 µg/100 g), oranges (260 µg/100 g), coffee (29 µg/100 g), and milk (18 µg/100 g) [249,250,251].

Given this background, a supplementation of 3 mg/day of boron can be considered useful for the support of bone health and the maintenance of an adequate BMD.

### 4.21. Selenium

A review by Zeng summarized the in-vitro and animal model studies concerning the molecular functions of Se relevant to bone health that involve the functions of selenoproteins that are antioxidant enzymes that participate in maintaining cell redox balance, which is important in the regulation of inflammation and bone cell proliferation/differentiation [252].

A subsequent review by Zhang confirmed that the presence of selenoproteins in bone is essential for normal skeletal development [253]. A study in an animal model highlighted how a selenium deficiency could retard growth, reduce bone density, and increase its resorption [254]. A study in humans demonstrated that the serum selenoprotein level correlated positively with total BMD and the serum selenium level with the total and femoral neck BMD in 387 aging men from Europe. It is vital to note that the observed relationships were independent of the thyroid function [255]; however, in a previous case-control study in which the serum level of some trace elements was assessed, including selenium in a population of 107 postmenopausal women divided into three groups, based on the value of BMD, no different serum selenium concentrations were observed between the various groups [256].

These conflicting opinions led researchers from Central South University of Changsha in China to conduct a cross-sectional study involving a large sample of the Chinese population aged 40 and over (6267 both male and female participants), where it was found that participants with the lowest levels of dietary selenium intake have higher prevalence of osteoporosis in a dose-response fashion [257]. Selenium is obtained from food intake. The average daily dose needed varies from 15 µg in the newborn to 55 µg in an adult, reaching 70 µg during breastfeeding.

In summary, selenium appears to be a powerful antioxidant element with a protective effect on the skeleton by maintaining cellular redox balance and counteracting bone resorption, and it is therefore essential, in order to maintain adequate BMD, to cover the daily needs by taking in foods rich in selenium, such as bovine kidney (145 µg), tuna (112 µg), murmurs (102 µg), clams (65 µg), sardines (58 µg), lobster (54 µg), mussels (42 µg), bovine liver (42 µg), sheep liver (42 µg), and sole (36 µg) (referred to 100 g of food), while the amount of selenium present in foods of plant origin depends on the amount of selenium contained in the soil in which they were grown.

### 4.22. Water

Water can prove to be an excellent source of various minerals and therefore extremely useful in achieving adequate levels of intake of the same [258].

Evidence of greatest interest is reported in a prospective study on the Norwegian population aimed at investigating the association between calcium present in drinking water and hip fracture and whether other minerals present in the water can modify this association. The investigation focused on trace metals present in 429 aqueducts, which supplied 64% of the population in Norway and was geographically linked to the addresses of patients who had suffered hip fracture accidents from 1992 to 2000. A total of 5433 men and 13,493 women between the ages of 50 and 85 suffered a hip fracture in the period 1994–2000. A high level of calcium in drinking water was associated with a 15% lower risk of hip fracture in men, but no significant difference was found in women. There was interaction between calcium and copper on hip fracture risk in men; the association between calcium and hip fracture risk was stronger when the copper concentration in the water was high than when it was low. No similar changes in risk were found in women. In conclusion, there is an inverse association between calcium in drinking water and the risk of hip fracture in men; the association is stronger when the concentration of copper in the water is high [259].

This highlights the importance of the composition of the water and tends to emphasize how its consumption contributes to achieving the adequate supply of important minerals for the body [260].

A 2008 study analyzed the mineral content of 150 randomly selected European waters, evaluating the PRAL index (potential renal acid load) in each of them by calculating the amount of potassium, magnesium, calcium, and sodium. Therefore, correlations emerged between the PRAL index of a given water and its effect on bone health: waters containing high quantities of SO_4_ (and therefore with a positive PRAL index) can in fact stimulate a greater excretion of calcium, unlike waters with higher quantities of HCO_3_ (with negative PRAL index), which instead can bring benefits to the bone. The ideal water, in order for the bone to benefit from it, should therefore have high amounts of HCO_3_ and calcium but low levels of SO_4_, criteria which 12% of European waters met [261].

A randomized double-blind placebo-controlled study in 2005 evaluated the effects of consuming high-calcium mineral water (HCaMW) on biochemical indices of bone remodeling in postmenopausal women with low dietary intake of approximately. In the six-month duration of the mentioned study, 180 postmenopausal women were recruited with a dietary intake of Ca less than 700 mg/day, dividing them into two groups: the control group, which was administered 1 L of HCaMW (596 mg Ca/L) per day and the placebo group, who was given 1 L of low-calcium mineral water (10 mg/L). Changes from baseline in biochemical indices after six months consisted of a significant 14.1% reduction in serum PTH, osteocalcin (−8.6%), bone alkaline phosphatase (−11.5%), serum (−16, 3%), and urine (–13.0%) telopeptide C collagen type 1 in the HCaMW group compared to the placebo group, where all biochemical indices increased after six months. Thus, a daily supplement of 596 mg of Ca through the consumption of 1 L of HCaMW was able to lower serum PTH and bone turnover indices in postmenopausal women with low calcium intake, helping to mitigate calcium deficiency and reduce age-related bone loss in this population [262].

In conclusion, the consumption of bicarbonate-calcium mineral water can be considered an important source of calcium for the population and a great help in reducing bone loss, particularly in postmenopausal women, where the need for calcium increases due to the decrease of estrogen levels. It is crucial to remember that 1 L of calcium water provides the same amount of calcium as 300 mL of milk or yogurt but is not, unlike them, a caloric source.

### 4.23. Caffeine

Caffeine, or theine, is a natural alkaloid present in coffee, cocoa, tea, cola, guarana, and mate plants and in the drinks obtained from them.

It has been hypothesized that coffee intake can lead to a worsening of the calcium balance in humans. The 1986 study by Yeh J. carried out on a mouse model showed that the daily administration of caffeine for four weeks causes an alteration of the balance of calcium in those situations in which there was an impairment of the ability to absorb calcium. The study also speculated that such a situation may exist in elderly human subjects since they have a reduced ability to synthesize 1,25-(OH)_2_ D [263].

A subsequent observational study conducted by Kynast-Gales and Massey in 1994 found a correlation between caffeine consumption and increased urinary excretion of calcium and magnesium in a small sample of subjects with normal renal and hepatic function: 17 healthy individuals of both sexes with ages 17 to 41 were recruited and subjected to a caffeine withdrawal/consumption test conducted on two consecutive days, while subjects consumed a controlled diet containing on average 11.3 mmol of Ca and 12.7 mmol of Mg; on the first day, the patients did not consume caffeine, and on the second day, two doses of caffeine of 3 mg/kg of lean mass were administered at 7:00 and 10:00 in the morning; and on both days, tests were performed for detection of the salivary caffeine concentration and serial urine collections every 3 h. From the measurements thus carried out, it was possible to highlight a net increase in urinary Ca and Mg after the second dose of caffeine administered, but caffeine had no significant effect on the urinary excretion of calcium or magnesium between 16:00 and 1:00 in the morning, while between 1:00 and 4:00 in the morning, the urinary excretion of Ca and Mg decreased after intake. It emerged, therefore, that nocturnal compensatory renal conservation was insufficient to compensate for the mineral losses induced by morning caffeine, with a consequent net urinary increase in the 24 h of excretion of 0.32 mmol Ca and 0.16 mmol Mg [264].

Additionally, in 1994, the study conducted by Massey LK wanted to determine the effect of two weeks of withdrawal from caffeine on calcium and bone metabolism in women who habitually consumed caffeine and moderate/low amounts of calcium. Participants were 25 women, aged 39 to 76, who routinely consumed at least 200 mg of caffeine per day. Three days of dietary recordings and 24-hour urine collection and fasting blood sampling were performed for women were classified as low calcium users (414–584 mg per day) or moderate calcium users (662–1357 mg per day) based on the three days of dietary records. It was found that women in the low-calcium group had higher serum calcium levels after caffeine withdrawal, while there were no differences in the moderate-calcium group. Women in the low-calcium group also had lower serum bone isoenzyme alkaline phosphatase levels after caffeine withdrawal, while no significant changes in bone alkaline phosphatase were observed in moderate-consuming patients. The mean 24-hour urinary excretion of calcium decreased after caffeine withdrawal only in the moderate-calcium group. The author therefore concluded that abstinence from moderate caffeine intake increases ultra-filterable calcium and decreases bone alkaline phosphatase in elderly women who consume <600 mg of calcium per day [265].

#### Long-Term Caffeine Intake and Bone Metabolism

A prospective study was conducted by Hallström et al. in 2005 on a large sample of female subjects with osteoporotic fractures (3279) for a follow-up period of 10.3 years; the subjects were divided into five quintiles based on the average amount of caffeine consumed daily. The results showed a significant difference between the risk of fracture between caffeine users in the top quintile (<200 mg/day) and those belonging to the highest quintile (>330 mg/day), observing an HR of 1.2 (CI: 1.07–1.35 at 95%). However, in this study, it was found that women belonging to the quintile with the highest consumption of caffeine are also those with the lowest calcium intake through the diet [266].

Rapuri et al., (2001) instead wanted to conduct an observational study comparing both longitudinally (follow-up time: three years) and horizontally a group of 96 women aged between 65 and 77 years to highlight changes in BMD in relation to the average daily caffeine consumption and the vitamin D receptor polymorphisms (VDR), reaching the following conclusions: women with an intake >300 mg/day of caffeine compared to those with an intake <300 mg/day show a higher rate of bone density loss (1.9 ± 0.97% and 1.19 ± 1.08% respectively, *p* = 0.038), and women with high caffeine intake (>300 mg/day) with the tt genotype for VDR are at greater risk of bone density loss than those with the TT genotype (*p*-value = 0.054); T and t are two allelic forms of vdr obtained by cutting the gene with the restriction enzyme Taq-1. The T allele is the dominant and t the recessive [267].

The meta-analysis conducted by Lee et al. in 2014 aimed to review the literature to quantify the association between fracture risk and coffee consumption. Participants (253,000) in nine cohort and six case-control studies were included in the study based on fracture risk and caffeine consumption. From the analysis of the data, it emerged that in female subjects, the risk of incurring fractures was linearly correlated to coffee consumption (compared to women who did not consume coffee, women who consume two cups a day are exposed to a risk increased by 2%; the risk for those who consume eight increased by 54%). In male subjects, on the other hand, an inverse trend was observed: subjects with the highest coffee intake were subject to a 24% reduced risk of bone fractures [268].

In 2017, the extensive systematic review conducted by Wikoff et al., wanted to investigate more widely than the previous one the possible harmful effects of caffeine on health in various groups of the population (young people, adults, and pregnant women). From the screening of over 5000 studies on caffeine in the literature, 381 studies were selected that met the inclusion and exclusion criteria. The analysis concluded that in adult subjects, the consumption of up to a maximum of 400 mg of caffeine and <300 mg for pregnant women does not appear to have adverse effects on any of the macro-areas investigated in the study (acute toxicity, cardiovascular toxicity, effects on calcium and bone, effects on behavior and reproduction). In conclusion, the authors stated that a caffeine consumption of 2.5 mg/kg of body weight is believed to be harmless to health [269].

Finally, the systematic review by Doepker et al. (2018) took up the macro-areas of the previous review by Wikoff et al., reaching to validate the same cut-offs in terms of doses of caffeine not harmful to health (<400 mg/day for healthy adults, <300 mg/day for pregnant women, and 2.5 mg/kg bodyweight for children and adolescents) but adding further considerations regarding the ratio of caffeine consumption to bone mineralization alterations: the authors acknowledged that in many studies, the effect of caffeine on bone was not adequately taken into account in the “calcium intake” variable, resulting in the latter being a crucial confounding factor. Furthermore, according to the authors, taking into account the risk of fracture, most of the data demonstrated a lack of effect at levels below and well above (up to 760 mg/day) at the cut-off of 400 mg/day, with a moderate level of confidence, underlining that there is no greater risk of fractures in subjects with adequate calcium intake at these levels of caffeine consumption in most of the studies analyzed and suggesting the need to plan more controlled studies in order to better define the role of caffeine and its relationship with calcium metabolism [270].

Considering the evidence found in the literature, the role of caffeine intake in the depletion of bone mass remains univocal: with certainty, it is possible to state that caffeine has a calcium and phosphatic effect in the immediate (minutes and hours after administration), but this effect has not been correlated with sufficient strength with chronic diseases, such as osteoporosis; consequently, it is not possible to define moderate caffeine intake as a risk factor for the development of osteoporosis.

Studies that investigated long-term caffeine intake as a possible cause of BMD reduction often did not consider important confounding factors, such as daily calcium intake in the diet, while those of a supplementary nature and with controlled calcium intake instead are very small and indelible.

The dose of 400 mg/day find sample confirmation in the literature (numerous meta-analyzes agree) as a daily dose not harmful to health; consequently, considering that the average amount of caffeine for espresso varies between 80 and 100 mg, in agreement with scientific evidence, a consumption of 3–4 coffees per day is not harmful to the well-being of the bone.

### 4.24. Salt/Sodium

Although chlorine and sodium are necessary elements for the human body, in the current Western diet, they are consumed excessively through the use of salt.

In addition to the negative effects that an excess of salt consumption can have on the cardiovascular system and the risk of developing gastric and intestinal cancers, ref. [271] excessive salt consumption, defined >5 g per day [272], it would seem to have a relationship with the onset of bone problems [273]. In fact, high amounts of salt modify the metabolism of calcium by increasing its urinary excretion and promoting the risk of osteopenia and osteoporosis [274].

#### Sodium Chloride Intake and Bone Metabolism

A cross-sectional study carried out in 2011 on 102 women aged between 19 and 35 years showed a correlation between an excessive urinary excretion of salt (which reflects the high oral intake), evaluated on 24-h urine collection, and a reduction of BMD, evaluated with DXA. In particular, the following correlation was observed between urinary calcium and sodium levels (always evaluated over 24 h): for every 100 mmol (2300 mg) of sodium excreted, the urinary calcium excreted increased by 1.1 mmol (44 mg) [275].

In the 2006 narrative review, thanks to the results of the studies that directly evaluated sodium intake (calculated through the administration of food frequency questionnaires), it was stated that 2300 mg of sodium (equivalent to about 5.7 g of salt) introduced into the diet is equivalent to the loss, in the kidney, of about 40 mg of calcium [274].

In addition, a study published in 2015 analyzed data from participants in the Korea National Health and Nutrition Examination Survey (KNHANES) from 2008 to 2011, resulting in a sample of 2779 postmenopausal Korean women (mean age 62.7 years), in which was shown a correlation between high urinary sodium excretion and low BMD and high prevalence of osteoporosis in the lumbar spine (parameters measured with DXA) [276].

In a subsequent 2017 study conducted on data from the same sample of postmenopausal Korean women, there was a prevalence of cases of lumbar spine osteoporosis (assessed with DXA) among women who consumed ≥4001 mg of salt per day compared to those who consumed ≤2000 mg (intakes always assessed through food frequency questionnaires). On the other hand, for the femoral neck, the prevalence rates of osteoporosis were significantly higher in those who consumed ≥5001 mg of salt per day compared with those who consumed ≤4000 mg. This demonstrates that high salt consumption is strongly associated with the development of osteoporosis in postmenopausal women [277].

In order to better differentiate the relationship between excessive sodium intake and its effect on bone mass on pre- or postmenopausal women, another cross-sectional study was conducted on KNHANES 2008–2011 data in 2017, which selected a sample of 9526 women over the age of 18. By analyzing and eliminating the confounding factors present in the two groups of women (such as the daily calories, the grams of protein in the diet; the milligrams of Ca, P, K and Na taken; the habit of smoking; the consumption of alcohol; physical activity levels; vitamin D values; etc.), there was a negative correlation between sodium intake and BMD/BMC in postmenopausal women but not in premenopausal women. Furthermore, among postmenopausal women, those with a sodium intake >2000 mg per day were at greater risk of osteoporosis than those who consumed <2000 mg [278].

Finally, in a recent systematic review and meta-analysis in 2018, 718 articles were analyzed in order to define the relationship between sodium (taken with the diet and/or urinary) and BMD, BMC, and risk of osteoporosis. In particular, the meta-analysis selected nine studies, for a total of 39,065 subjects, which demonstrate how a high sodium consumption (calculated on the sodium taken daily, through questionnaires, or on that excreted in the urine in 24 h) significantly increases the risk of osteoporosis. To reduce the great heterogeneity between the studies, subgroups were created based, for example, on sex, the type of sodium considered (whether urinary or diet), and the age of the subjects. In particular, the analysis showed that the risk of osteoporosis is higher among menopausal women than in pre-menopausal women and in general in women over 50. Analyzing the correlations based on the type of sodium (urinary or food), it was noted that the sodium introduced with the diet is associated with the risk of osteoporosis, while not enough evidence was found to define a correlation between urinary sodium and the risk of osteoporosis. Equally, no significant correlation with BMD could be defined [273].

The correlation between sodium intake and risk of osteoporosis can be explained by the negative action that salt exerts on calcium metabolism [279]. In fact, sodium acts in the kidney by increasing the excretion of calcium and in this way can lead to an increase in remodeling and bone loss [273]. Furthermore, sodium also acts on the metabolism of hydroxyproline (amino acid derivative contained in collagen) and parathyroid hormone (involved in calcium metabolism), also contributing in this way to bone remodeling [280]. It is therefore obvious how the calcium level can be a confounding factor, and a limitation for these studies that must be taken into consideration.

In conclusion, although there are studies that have given conflicting results, the evidence gathered confirms the idea that it is advisable to reduce the intake of salt in one’s diet: values of salt intake <2000 mg per day allow protection against bone loss.

In fact, high amounts of salt modify the metabolism of calcium by increasing its urinary excretion and promoting the risk of osteopenia and osteoporosis.

It is therefore advisable to limit the consumption of those foods rich in salt, such as preserved and canned foods, packaged ones, processed meat, and aged cheeses. In the kitchen, to flavor your dishes, you can freely use spices and aromatic herbs so as to further reduce the use of salt.

### 4.25. Isoflavones

Isoflavones are a type of vegetal-derived polyphenol found in a limited variety of plants (in particular in *Leguminosae* and *Iridaceae* families). The primary isoflavones are genistein, daidzein, and glycitein, and the richest alimentary source of isoflavones are soybeans and all soybean-derived foods (Tofu, tempeh) although in lesser quantities than the bean itself [281].

The intake of isoflavones and equol (a primary product of daidzein metabolism by the gut microflora) has been associated with various health benefits due to their anti-inflammatory [282], anti-oxidant [283], and hormone-like activities [284].

A recent narrative review [46] showed that isoflavones may play a significant role in preventing BMD loss: isoflavones (in particular daidzein) have showed direct effect in promoting osteoblast differentiation [285] and in suppressing the cellular expression of RANK-L [286] in in-vitro studies; also, a variety of human clinical trials assessed the potential benefit of the intake of flavonoids in osteopenia/osteoporosis and found possible positive correlation between flavonoid supplementation and attenuation of BMD loss [287,288,289].

In particular a relevant randomized double-blind placebo-controlled trial regarding the effect of flavonoids on BMD [290] in 389 post-menopausal women randomly assigned to treatment group (supplementation of 54 mg of genistein daily) and a placebo group demonstrated a significative BMD difference in both lumbar spine and femoral neck after 24 months between treatment and control group, showing a positive effect of genistein on bone health in elderly women.

In conclusion, according to these recent evidences isoflavones intake may play a significant role in osteopenia/osteoporosis prevention and treatment although more evidences are needed.

## 5. Conclusions

In conclusion, because bone is a nutritionally modulated tissue, a balanced nutritional intake of specific nutrients (from a specific diet and/or through a dietary integration) can be considered as the first step for an effective preventive strategy against an osteopenic state. Therefore, this narrative review was intended to evaluate the latest data regarding ideal dietary approach in order to reduce BMD loss and to construct a food pyramid that allows osteopenia/osteoporosis patients to easily figure out what to eat. The results of the review demonstrated, as shown graphically in the pyramid in the Figure 2, that carbohydrates should be consumed every day (three portions of whole grains for fiber and zinc content), together with fruits and vegetables (five portions; in particular, orange-colored fruits and vegetables due to their beta-carotene and vitamin C content and green leafy vegetables for vitamin K, folic acid, magnesium, and prebiotics content are preferred), light yogurt (125 mL), skim milk (200 mL), extra virgin olive oil (almost 20 mg/day for vitamin E and polyphenols content), and calcium water (almost 1 L/day also for stone-forming subjects); weekly portions should include fish (four portions for vitamin D, proteins and omega 3 content, considering EPA + DHA at least 0.52 g/day), white meat (three portions for proteins, iron, and vitamin B12 content), legumes (two portions for vegetal proteins), eggs (two portions), cheeses (two portions for proteins, calcium, vitamin B6), and red or processed meats (once/week). At the top of the pyramid, there are two pennants: one green means that osteopenia/osteoporosis subjects need some personalized supplementation (if daily requirements cannot be satisfied through diet, calcium, vitamin D, boron, omega 3, and isoflavones supplementation could be an effective strategy with a great benefit/cost ratio), and one red means that there are some foods that are banned (salt, sugar, and inorganic phosphate additives). Finally, three to four/times/week, 30–40 min of aerobic and resistance exercises must be performed.

From the authors’ viewpoint, future research and development could include several investigation topics where limited scientific evidence is present and that are particularly focused, such as (1) the impact of gut microbiota composition on BMD also considering the effectiveness of prebiotics and probiotics supplementation on BMD and (2) the effectiveness of polyphenol intake contained in fruit, vegetables, EVO oil, and soybeans (assessed by food frequency questionnaires) on BMD.

## Figures and Tables

**Figure 1 nutrients-14-00074-f001:**
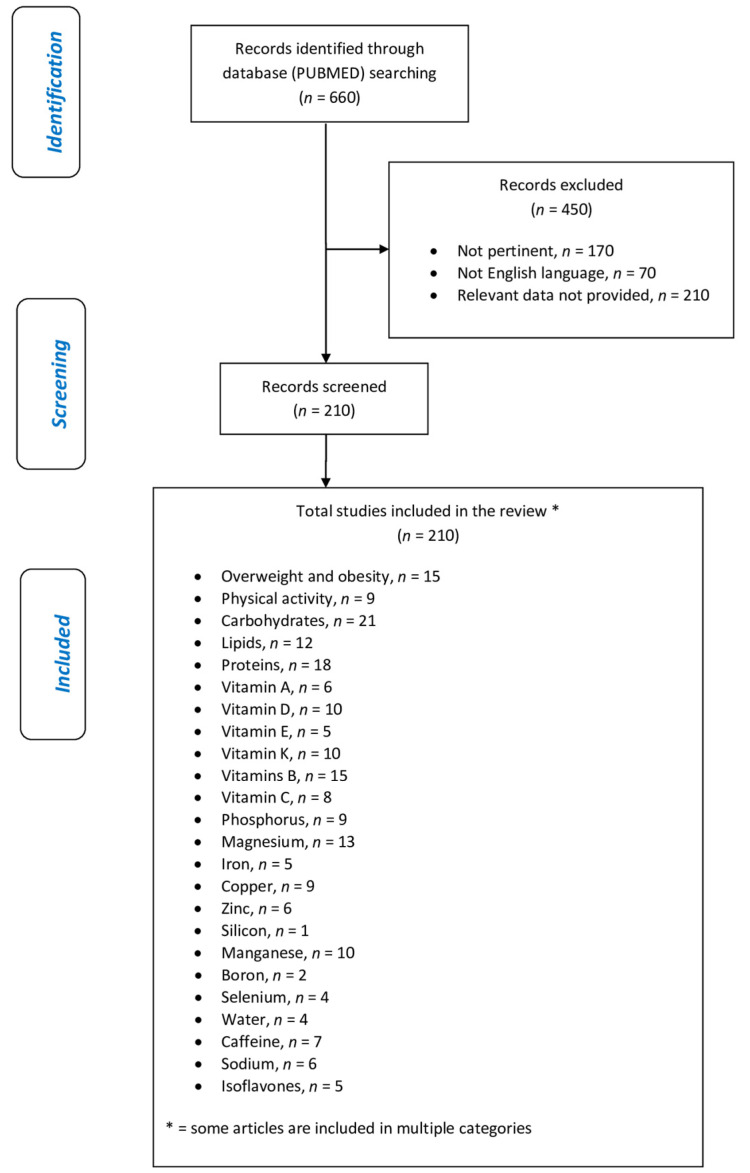
Flow chart of the literature research.

**Figure 2 nutrients-14-00074-f002:**
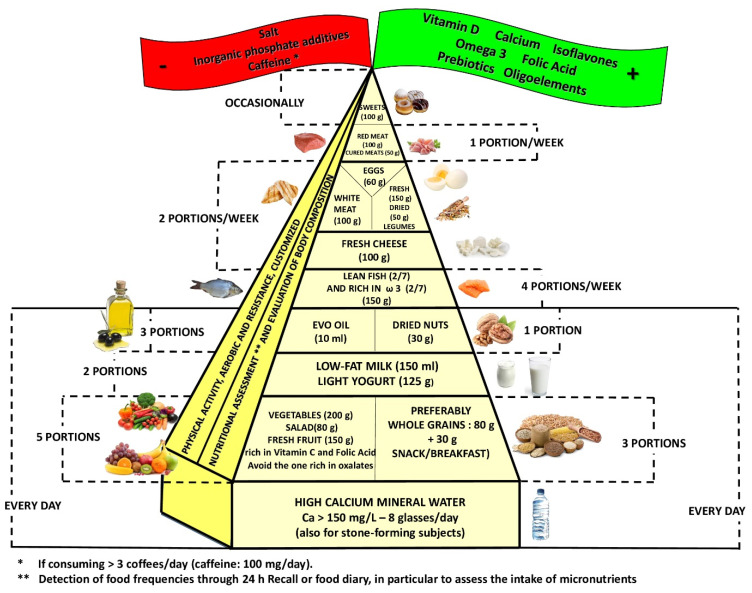
Ideal food pyramid for patients with osteopenia/osteoporosis. “+”, one green pennant, means that osteopenia/osteoporosis subjects need some personalized supplementation (if daily requirements cannot be satisfied through diet, calcium, vitamin D, boron, omega 3, and isoflavones supplementation could be an effective strategy with a great benefit/cost ra-tio); “-”, one red pennant, means that there are some foods that are banned (salt, sugar, and in-organic phosphate additives). Finally, three to four/times/week, 30–40 min of aerobic and resistance exercises must be performed.

## Data Availability

The data presented in this study are available in the article and in the Appendix A.

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
