# Peer review of "Nutrition, Physical Activity, and Dietary Supplementation to Prevent Bone Mineral Density Loss: A Food Pyramid"

_nutrients, 2021, doi:10.3390/nu14010074_

Round 1

Reviewer 1 Report

The authors wrote an interesting and comprehensive narrative review on the evidence regarding the ideal dietary therapy in order to reduce the risk of loss of bone mineral density and to construct a food pyramid for osteopenic/osteoporotic individuals.

The studied data were obtained in a well-organised manner and key references were included. Methods are comprehensively described. The authors based their review on the strength of the evidence from clinical nutrition supported by a variety of studies, including evidence from a systematic review or meta-analysis of relevant randomized controlled trials or evidence-based clinical practice guidelines based on systematic reviews of randomized controlled trials as well as evidence obtained from at least one well-designed randomized controlled trial (e.g. large multi-site randomized controlled trial) or without randomization or case-control or cohort studies, and prospective longitudinal studies.

It should be noted that the authors propose practical dietary recommendations for osteopenic/osteoporotic individuals graphically in the form of a food pyramid.

This work focuses on an interesting area but the manuscript is descriptive in nature and the results are associative and confirmatory, to some extent, of previous data. To improve the manuscript, please adjust the section Conclusions to identify directions for further research where limited scientific evidence is present.

Author Response

the section conclusions has been revised according to your suggestions

Reviewer 2 Report

In the present review, Mariangela Rondanelli and colleagues evaluated the latest date regarding ideal dietary approach in order to reduce bone mineral density loss and to construct a food pyramid that allows osteopenia/osteoporosis patients to easily figure out what to eat. The authors concluded that at the top of the pyramid, there are two pennants: one green means that osteopenia/osteoporosis subjects need some personalized supplementation (if daily requirements cannot be satisfied through diet, calcium, vitamin D, boron, omega 3 supplementation could be an effective strategy with a great benefit/cost ratio) and one red means that there are some foods that 30 are banned (salt, sugar, inorganic phosphate additives). Finally, three-four/times/week 30-40 minutes of aerobic and resistance exercises must be performed.

Overall, I think that the paper is well written, nice, and timely, and it will be of great interest to the readers and researchers, in general.

I make some suggestions for further improve the quality of the manuscript.

1) “Oriental diet” (similar to “Mediterranean diet”) is particularly abundant in isoflavones. Specifically, the positive effects on osteopenic, postmenopausal women could be related to these phytochemicals.

2) Recent research suggested that dietary fibers from beans, fruits and vegetables were associated with the gut microbiome composition and, accordingly, with bone diseases.

Please discuss these intriguing aspects in the manuscript considering for your convenience these references (Harahap I.A and Suliburska J. Foods. 2021;10(11):2685; Marini H. e t al. Ann Intern Med. 2007;146(12):839-47; Marini H. et al. J Bone Miner Res. ;23(5):715-20).

Author Response

1) a new paragraph, entitled “isoflavones”, has been added in order to develop this topic

2) new sentences, with references suggested, have been added in the paragraph entitled "carbohydrates
